# Towards Interpretable Mental Health Analysis with Large Language Models

**Kailai Yang** [1]    **Shaoxiong Ji** [*2]    **Tianlin Zhang** [*1] **Qianqian Xie** [†1]
**Ziyan Kuang** [4]    **Sophia Ananiadou** [1,3]
[1] The University of Manchester    [2] University of Helsinki
[3] Artificial Intelligence Research Center, AIST    [4] Jiangxi Normal University
{kailai.yang,tianlin.zhang}@postgrad.manchester.ac.uk
shaoxiong.ji@helsinki.fi; {xqq.sincere,plumjane1225}@gmail.com
sophia.ananiadou@manchester.ac.uk

## Abstract

The latest large language models (LLMs) such as ChatGPT, exhibit strong capabilities in automated mental health analysis. However, existing relevant studies bear several limitations, including inadequate evaluations, lack of prompting strategies, and ignorance of exploring LLMs for explainability. To bridge these gaps, we comprehensively evaluate the mental health analysis and emotional reasoning ability of LLMs on 11 datasets across 5 tasks. We explore the effects of different prompting strategies with unsupervised and distantly supervised emotional information. Based on these prompts, we explore LLMs for interpretable mental health analysis by instructing them to generate explanations for each of their decisions. We convey strict human evaluations to assess the quality of the generated explanations, leading to a novel dataset with 163 human-assessed explanations[1]. We benchmark existing automatic evaluation metrics on this dataset to guide future related works. According to the results, ChatGPT shows strong in-context learning ability but still has a significant gap with advanced task-specific methods. Careful prompt engineering with emotional cues and expert-written few-shot examples can also effectively improve performance on mental health analysis. In addition, ChatGPT generates explanations that approach human performance, showing its great potential in explainable mental health analysis.

## 1 Introduction

WARNING: This paper contains examples and descriptions that are depressive in nature.

Mental health conditions such as depression and suicidal ideation seriously challenge global health care (Zhang et al., 2022a). NLP researchers have devoted much effort to automatic mental health analysis, with current mainstream methods leveraging the Pre-trained Language Models (PLMs) (Yang et al., 2022; Abed-Esfahani et al., 2019). Most recently Large Language Models (LLMs) (Brown et al., 2020; Ouyang et al., 2022), especially ChatGPT [2] and GPT-4 (OpenAI, 2023), have exhibited strong general language processing ability (Wei et al., 2022; Luo et al., 2023; Yuan et al., 2023). In mental health analysis, Lamichhane (2023) evaluated ChatGPT on stress, depression, and suicide detection and glimpsed its strong language understanding ability to mental health-related texts. Amin et al. (2023) compared the zero-shot performance of ChatGPT on suicide and depression detection with previous fine-tuning-based methods.

Though previous works depict a promising future for a new LLM-based paradigm in mental health analysis, several issues remain unresolved. Firstly, mental health condition detection is a safe-critical task requiring careful evaluation and high transparency for any predictions (Zhang et al., 2022a), while these works simply tested on a few binary mental health condition detection tasks and lack the explainability on detection results. Moreover, other important mental health analysis tasks, such as the cause/factor detection of mental health conditions (Mauriello et al., 2021; Garg et al., 2022), were ignored. Secondly, previous works mostly use simple prompts to detect mental health conditions directly. These vanilla methods ignore useful information, especially emotional cues, which are widely utilized for mental health analysis in previous works (Zhang et al., 2023). We believe it requires a comprehensive exploration and evaluation of the ability and explainability of LLMs on mental health analysis, including mental health detection, emotional reasoning, and cause detection

---

[*] Equal contribution, listed alphabetically.

[†] Corresponding author. Qianqian is now affiliated with Yale University. The work was done when she was at The University of Manchester.

[1]The data is released at https://github.com/SteveKGYang/MentalLLaMA

[2]https://openai.com/blog/chatgpt

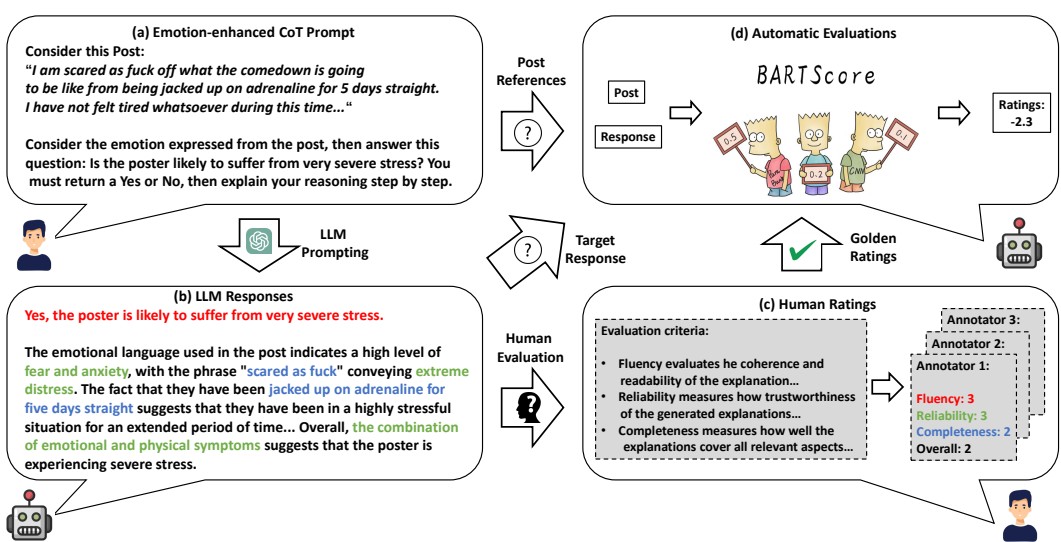

Figure 1: The pipeline of obtaining and evaluating the LLM-generated explanations for mental health analysis. In LLM responses, red, green, and blue words are marked as relevant clues for rating fluency, reliability, and completeness in human evaluations.

of mental health conditions. Therefore, we raise the following three research questions (RQ):

- **RQ 1**: How well can LLMs perform in generalized mental health analysis and emotional reasoning with zero-shot/few-shot settings?

- **RQ 2**: How do different prompting strategies and emotional cues impact the mental health analysis ability of ChatGPT?

- **RQ 3**: How well can ChatGPT generate explanations for its decisions on mental health analysis?

To respond to these research questions, we first conduct a preliminary study of how LLMs perform on mental health analysis and emotional reasoning. We evaluate four LLMs with varying model sizes including ChatGPT, InstructGPT-3 (Ouyang et al., 2022), LLaMA-13B, and LLaMA-7B (Touvron et al., 2023), on 11 datasets across 5 tasks including binary/multi-class mental health condition detection, cause/factor detection of mental health conditions, emotion recognition in conversations, and causal emotion entailment. We then delve into the effectiveness of different prompting strategies on mental health analysis, including zero-shot prompting, Chain-of-Thought (CoT) prompting (Kojima et al., 2022), emotion-enhanced prompting, and few-shot emotion-enhanced prompting. Finally, we explore how LLMs perform for interpretable mental health analysis, where we instruct two representative LLMs: ChatGPT and InstructGPT-3,

to generate natural language explanations for each of its results on mental health analysis. To assess the quality of LLMs-generated explanations, we perform human evaluations by following a strict annotation protocol designed by domain experts, and thus create the novel dataset with 163 human-assessed explanations of posts from LLMs, aimed at facilitating the investigating of explainable mental health analysis methods and automatic evaluation metrics. We benchmark numerous existing automatic evaluation metrics on the corpus to guide future research on automatically evaluating explainable mental health analysis. We conclude our findings as follows:

1) **Overall Performance.** ChatGPT achieves the best performance among all examined LLMs, although it still significantly underperforms advanced supervised methods, highlighting the challenges of emotion-related subjective tasks.

2) **Prompting Strategies.** While a simple CoT trigger sentence is ineffective for mental health analysis, ChatGPT with unsupervised emotion-enhanced CoT prompts achieves the best performance, showing the importance of prompt engineering in leveraging emotional cues for mental health analysis. Few-shot learning from expert-written examples also significantly improves model performance.

3) **Explainability.** ChatGPT can generate approaching-human explanations for its classifications, indicating its potential to enhance the trans-

parency of mental health analysis. Current best automatic evaluation metrics can moderately correlate with human evaluations, indicating the need for developing customized automatic evaluation methods in explainable mental health analysis.

4) **Limitations.** Although its great potential, ChatGPT bears limitations on inaccurate reasoning and unstable predictions caused by its excessive sensitivity to minor alterations in prompts, inspiring future directions on improving ChatGPT and prompts. Unstable prediction problems can be mitigated by few-shot learning.

Our contributions can be summarized as follows: 1) We evaluate four representative LLMs on mental health analysis, 2) We investigate the effectiveness of prompting strategies including CoT, emotion-enhanced prompts, and few-shot learning for mental health analysis, 3) We explore LLMs for explainable mental health analysis, and conduct human and automatic evaluations on LLMs-generated explanations, 4) We create the first evaluation dataset with LLMs-generated explanations rigorously assessed by domain experts, for examining and developing of automatic evaluation metrics, 5) We analyze the potential and limitations of LLMs and different prompting strategies for mental health analysis.

## 2 Methodology

This section introduces the details of evaluated LLMs and different prompting strategies for improving LLMs' efficiency and explainability in mental health analysis. Due to the page limits, all evaluations, experiments, and analyses on emotional reasoning are presented in Appendix B. We also perform human evaluations on the quality of LLM-generated explanations and benchmark existing automatic evaluation metrics on the human evaluation results, where an example is shown in Figure 1.

### 2.1 Large Language Models

We benchmark the following powerful LLMs for the zero-shot mental health analysis:

1) **LLaMA-7B/13B.** LLaMA (Touvron et al., 2023) is a set of open-source LLMs developed by Meta AI, which are generatively pre-trained on entirely publicly available datasets. We test the zero-shot mental health analysis tasks on LLaMA models with 7 billion (LLaMA-7B) and 13 billion (LLaMA-13B) parameters.

2) **InstructGPT-3.** InstructGPT-3 continually trains GPT-3 (Brown et al., 2020) with instruction tuning (Ouyang et al., 2022), which enables the model to solve tasks in a question-answering format. We utilize *curie-instruct-beta* version (13 billion parameters) in our experiments.

3) **ChatGPT.** ChatGPT (*gpt-3.5-turbo*) is trained based on the 175 billion parameters version of InstructGPT (Ouyang et al., 2022) and continually optimized through reinforcement learning from human feedback (RLHF) (Stiennon et al., 2020).

### 2.2 In-context Learning as Explainable Mental Health Analyzer

In-context learning (Brown et al., 2020) elicits the powerful ability of LLMs given the information provided in the context without explicit updates of model parameters. We instruct LLMs with task-specific instructions to trigger their ability as the zero-shot analyzer for different mental health analysis tasks. We systematically explore three different prompting strategies for mental health analysis, i.e., straightforward zero-shot prompting with natural language query, emotion-enhanced Chain-of-Thought (CoT) (Wei et al., 2022), and distantly supervised emotion-enhanced instructions. The straightforward zero-shot prompting guides four LLMs by asking for a classification result from their responses. For example, for binary mental health condition detection, we design the following prompt:

*Post: "[Post]". Consider this post to answer the question: Is the poster likely to suffer from very severe [Condition]? Only return Yes or No.*

**Emotion-enhanced Prompts** Moreover, we design three emotion-enhanced prompting strategies to better instruct ChatGPT to conduct explainable mental health analysis: 1) **Emotion-enhanced CoT prompting**. We perform emotion infusion by designing unsupervised emotion-enhanced zero-shot CoT prompts, where the emotion-related part inspires the LLM to concentrate on the emotional clues from the post, and the CoT part guides the LLM to **generate step-by-step explanations** for its decision. This improves the explainability of LLMs' performance. For example, for the binary detection task, we modify the zero-shot prompt as follows:

*Post: "[Post]". Consider the emotions expressed from this post to answer the question:*

*Is the poster likely to suffer from very severe [Condition]? Only return Yes or No, then explain your reasoning step by step.*

2) **Supervised emotion-enhanced prompting**. In addition, we propose a distantly supervised emotion fusion method by using sentiment and emotion lexicons. We utilize the VADER (Hutto and Gilbert, 2014) and NRC EmoLex (Mohammad and Turney, 2010, 2013) lexicons to assign a sentiment/emotion score to each post and convert the score to sentiment/emotion labels. Then we design emotion-enhanced prompts by adding the sentiment/emotion labels to the proper positions of the zero-shot prompt. 3) **Few-shot Emotion-enhanced Prompts**. We further evaluate the impact of few-shot examples on emotion-enhanced prompts. We invite domain experts (Ph.D. students majoring in quantitative psychology) to write one response example for each label class within a test set, where all responses consist of a prediction and an explanation describing the rationale behind the decision. We then include these examples in the emotion-enhanced prompts to enable in-context learning of the models. For example, for the binary detection task, we modify the original emotion-enhanced prompt to combine $N$ expert-written explanations in a unified manner:

*You will be presented with a post. Consider the emotions expressed in this post to identify whether the poster suffers from [condition]. Only return Yes or No, then explain your reasoning step by step. Here are N examples:*
*Post: [example 1]*
*Response: [response 1]*
*...*
*Post: [example N]*
*Response: [response N]*

*Post: [Post]*
*Response:*

**Task-specific Instructions**  We conduct broad tests of LLMs' mental health analysis ability on the following three tasks: binary mental health condition detection, multi-class mental health condition detection, and cause/factor detection of mental health conditions. Binary mental health condition detection is modeled as a yes/no classification of the mental health condition, such as depression and stress from a post. In contrast, multi-class detection identifies one label from multiple mental health conditions. Cause/factor detection aims at

recognizing one potential cause of a mental health condition from multiple causes. More details about the prompt design and examples of the prompts are presented in Appendix C.2.

### 2.3 Evaluation for Explainability

**Human Evaluation**  We examine the quality of the generated explanations by two representative LLMs: ChatGPT and InstructGPT-3, with human evaluations on the binary mental health conditions detection task. We utilize ChatGPT and InstructGPT-3 to simultaneously generate explanations for the same posts with the same emotion-enhanced CoT prompts. The annotation protocol is developed through collaborative efforts with 2 domain experts (Ph.D. students majoring in quantitative psychology) and considerations of human evaluation criteria for other text generation tasks (Wallace et al., 2021; DeYoung et al., 2021). Specifically, four key aspects are assessed: 1) **Fluency**: the coherence and readability of the explanation. 2) **Reliability**: the trustworthiness of the generated explanations to support the prediction results. 3) **Completeness**: how well the generated explanations cover all relevant aspects of the original post. 4) **Overall**: the general effectiveness of the generated explanation.

Each aspect is divided into four standards rating from 0 to 3. Higher ratings reflect more satisfactory performance and 3 denotes approaching human performance. Each LLM-generated explanation is assigned a score by 3 annotators for each corresponding aspect, followed by the examination of 1 domain expert. All annotators are PhD students with high fluency in English. We evaluate 121 posts that are correctly classified by both ChatGPT (ChatGPT$_{true}$) and InstructGPT-3 to enable fair comparisons. 42 posts that are incorrectly classified by ChatGPT (ChatGPT$_{false}$) are also collected for error analysis and examination of the automatic evaluation metrics. We will release the annotated corpus for facilitating future research. Details of the criteria are described in Appendix E.

**Automatic Evaluation**  Though human evaluations provide an accurate and comprehensive view of the generated explanations' quality, they require huge human efforts, making it hard to be extended to large-scale datasets. Therefore, we explore utilizing automatic evaluation metrics, originally developed for generation tasks such as text summarization, to benchmark the evaluation on

our annotated corpus. We rely on the ability of the evaluation models to score the fluency, reliability, and completeness of the explanations. We select the following widely utilized metrics to automatically evaluate LLM-generated explanations: BLEU (Papineni et al., 2002), ROUGE-1, ROUGE-2, ROUGE-L (Lin, 2004), GPT3-Score (Fu et al., 2023) (*davinci-003*), and BART-Score (Yuan et al., 2021). We also use the BERT-score-based (Zhang et al., 2020) methods with different PLMs, including the domain-specific PLMs MentalBERT and MentalRoBERTa (Ji et al., 2022b), except for BERT and RoBERTa.

## 3 Experimental Settings

**Mental Health Analysis** Firstly, we introduce the benchmark datasets, baseline models, and automatic evaluation metrics for the classification results of mental health analysis.

**Datasets.** For binary mental health condition detection, we select two depression detection datasets Depression_Reddit (DR) (Pirina and Çöltekin, 2018), CLPsych15 (Coppersmith et al., 2015), and another stress detection dataset Dreaddit (Turcan and McKeown, 2019). For multi-class mental health condition detection, we utilize the dataset T-SID (Ji et al., 2022a). For cause/factor detection of mental health conditions, we use a stress cause detection dataset called SAD (Mauriello et al., 2021) and a depression/suicide cause detection dataset CAMS (Garg et al., 2022). More details of these datasets are presented in Table 8 in the appendix.

**Baseline Models.** We select the following baseline models: CNN (Kim, 2014), GRU (Cho et al., 2014), BiLSTM_Att (Zhou et al., 2016), fast-Text (Joulin et al., 2017), BERT/RoBERTa (Devlin et al., 2019; Liu et al., 2019), and Mental-BERT/MentalRoBERTa (Ji et al., 2022b). Details about these baseline models are in Appendix D.2.

**Metrics.** We evaluate the model performance using the recall and weighted-F1 scores as the evaluation metric for all mental health datasets. Due to imbalanced classes in some datasets such as DR, CLPsych15, and T-SID, we use weighted-F1 scores following previous methods. In addition, it is crucial to minimize false negatives, which refers to cases where the model fails to identify individuals with mental disorders. Therefore, we also report the recall scores.

**Evaluation for Explainability** For the human evaluation results, we evaluate the quality of the annotations by calculating the inter-evaluator agreement: Fleiss' Kappa statistics (Fleiss et al., 2013) for each aspect. Any annotations with a majority vote are considered as reaching an agreement. To compare the automatic evaluation methods, we also compute Pearson's correlation coefficients between the automatic evaluation results and the human evaluation results, where higher values reflect more linear correlations between the two sets of data.

## 4 Results and Analysis

We conduct all LLaMA experiments on a single Nvidia Tesla A100 GPU with 80GB of memory. InstructGPT-3 and ChatGPT results are obtained via the OpenAI API. Each prompt is fed independently to avoid the effects of dialogue history.

### 4.1 Mental Health Analysis

The experimental results of mental health analysis are presented in Table 1. We first compare the zero-shot results of LLMs to gain a straight view of their potential in mental health analysis, then analyze ChatGPT's performance with other prompts enhanced by emotional information.

**Zero-shot Prompting.** In the comparison of LLMs, ChatGPT significantly outperforms LLaMA-7B/13B and InstructGPT-3 on all datasets. LLaMA-7B$_{ZS}$ displays random-guessing performance on multi-class detection (T-SID) and cause detection (SAD, CAMS), showing its inability to perform these more complex tasks. With an expanded model size, LLaMA-13B$_{ZS}$ achieves no better performance than LLaMA-7B. Though trained with instruction tuning, InstructGPT-3$_{ZS}$ still does not improve performance, possibly because the model size limits the LLM's learning ability. Compared with supervised methods, ChatGPT$_{ZS}$ significantly outperforms traditional light-weighted neural network-based methods such as CNN and GRU on binary detection and cause/factor detection, showing its potential in cause analysis for mental health-related texts. However, ChatGPT$_{ZS}$ struggles to achieve comparable performance to fine-tuning methods such as MentalBERT and MentalRoBERTa. Particularly, ChatGPT$_{ZS}$ achieves much worse performance than all baselines on T-SID. We notice that T-SID collects mostly short posts from Twitter with many

| Model | DR | | CLPsych15 | | Dreaddit | | T-SID | | SAD | | CAMS | |
|---|---|---|---|---|---|---|---|---|---|---|---|---|
| | Rec. | F1 | Rec. | F1 | Rec. | F1 | Rec. | F1 | Rec. | F1 | Rec. | F1 |
| **Supervised Methods** | | | | | | | | | | | | |
| CNN | 80.54 | 79.78 | 51.67 | 40.28 | 65.31 | 64.99 | 71.88 | 71.77 | 39.71 | 38.45 | 36.26 | 34.63 |
| GRU | 61.72 | 62.13 | 50.00 | 46.76 | 55.52 | 54.92 | 67.50 | 67.35 | 35.91 | 34.79 | 34.19 | 29.33 |
| BiLSTM_Att | 79.56 | 79.41 | 51.33 | 39.20 | 63.22 | 62.88 | 66.04 | 65.77 | 37.23 | 38.50 | 34.98 | 29.49 |
| fastText | 83.99 | 83.94 | 58.00 | 56.48 | 66.99 | 66.92 | 69.17 | 69.09 | 38.98 | 38.32 | 40.10 | 34.92 |
| BERT | 91.13 | 90.90 | 64.67 | 62.75 | 78.46 | 78.26 | 88.44 | 88.51 | 62.77 | 62.72 | 40.26 | 34.92 |
| RoBERTa | **95.07** | **95.11** | 67.67 | 66.07 | 80.56 | 80.56 | 88.75 | 88.76 | 66.86 | 67.53 | 41.18 | 36.54 |
| MentalBERT | 94.58 | 94.62 | 64.67 | 62.63 | 80.28 | 80.04 | 88.65 | 88.61 | 67.45 | 67.34 | 45.69 | 39.73 |
| MentalRoBERTa | 94.33 | 94.23 | **70.33** | **69.71** | **81.82** | **81.76** | **88.96** | **89.01** | **68.61** | **68.44** | **50.48** | **47.62** |
| **Zero-shot LLM-based Methods** | | | | | | | | | | | | |
| LLaMA-7B$_{ZS}$ | 63.55 | 58.91 | 57.0 | 56.26 | 54.83 | 53.51 | 23.04 | 25.55 | 10.53 | 11.04 | 13.92 | 16.34 |
| LLaMA-13B$_{ZS}$ | 67.24 | 54.07 | 50.0 | 39.29 | 47.83 | 36.28 | 23.04 | 25.27 | 12.57 | 13.2 | 13.12 | 14.64 |
| InstructGPT-3$_{ZS}$ | 58.87 | 58.66 | 50.33 | 49.86 | 50.07 | 49.88 | 27.60 | 26.27 | 12.70 | 9.36 | 10.70 | 12.23 |
| ChatGPT$_{ZS}$ | 82.76 | 82.41 | 60.33 | 56.31 | 72.72 | 71.79 | 39.79 | 33.30 | 55.91 | 54.05 | 32.43 | 33.85 |
| **Emotion-enhanced CoT LLM-based Methods** | | | | | | | | | | | | |
| ChatGPT$_{V}$ | 79.51 | 78.01 | 59.20 | 56.34 | 74.23 | 73.99 | 40.04 | 33.38 | 52.49 | 50.29 | 28.48 | 29.00 |
| ChatGPT$_{N\_sen}$ | 80.00 | 78.86 | 58.19 | 55.50 | 70.87 | 70.21 | 39.00 | 32.02 | 52.92 | 51.38 | 26.88 | 27.22 |
| ChatGPT$_{N\_emo}$ | 79.51 | 78.41 | 58.19 | 53.87 | 73.25 | 73.08 | 39.00 | 32.25 | 54.82 | 52.57 | 35.20 | 35.11 |
| ChatGPT$_{CoT}$ | 82.72 | 82.9 | 56.19 | 50.47 | 70.97 | 70.87 | 37.66 | 32.89 | 55.18 | 52.92 | 39.19 | 38.76 |
| ChatGPT$_{CoT\_emo}$ | 83.17 | 83.10 | 61.41 | 58.24 | 75.07 | 74.83 | 34.76 | 27.71 | 58.31 | 56.68 | 43.11 | 42.29 |
| ChatGPT$_{CoT\_emo\_FS}$ | **85.73** | **84.22** | **63.93** | **61.63** | **77.80** | **75.38** | **49.03** | **43.95** | **66.05** | **63.56** | **48.75** | **45.99** |

Table 1: Test results on the mental health analysis tasks. ChatGPT$_V$, ChatGPT$_{N\_sen}$, and ChatGPT$_{N\_emo}$ denote the emotion-enhanced prompting methods with VADER sentiments, NRC EmoLex sentiments, and NRC EmoLex emotions. ChatGPT$_{CoT}$ and ChatGPT$_{CoT\_emo}$ denote the zero-shot and emotion-enhanced CoT methods on the corresponding task. ChatGPT$_{CoT\_emo\_FS}$ combines expert-written few-shot examples in the emotion-enhanced prompt. The results of baseline methods are referenced from (Ji et al., 2022b).

usernames, hashtags, and slang words. The huge gap between the posts and ChatGPT's training data can make zero-shot detection difficult (Kocoń et al., 2023). Moreover, although the zero-shot CoT prompting is proven to be effective on most NLP tasks (Zhong et al., 2023; Wei et al., 2022; Kojima et al., 2022), we surprisingly find that ChatGPT$_{CoT}$ has a comparable or even worse performance with ChatGPT$_{ZS}$. This illustrates that the simple CoT trigger sentence is not effective in mental health analysis. Overall, ChatGPT significantly outperforms other LLMs, and exhibited some generalized ability for mental health analysis. However, it still underperforms fine-tuning-based methods, leaving a huge gap in further exploring LLMs' mental health analysis ability.

**Emotion-enhanced Prompting.** We further test ChatGPT with emotion-enhanced prompts on all datasets. Firstly, with the sentiment information from the lexicon VADER and NRC EmoLex, we notice that ChatGPT$_V$ and ChatGPT$_{N\_sen}$ perform worse than ChatGPT$_{ZS}$ on most datasets, showing that these prompts are not effective in enhancing model performance. A possible reason is that the coarse-grained sentiment classifications based on the two lexicons cannot describe complex emotions

expressed in the posts. Therefore, we incorporate fine-grained emotion labels from NRC EmoLex into the zero-shot prompt. The results show that ChatGPT$_{N\_emo}$ outperforms ChatGPT$_{N\_sen}$ on most datasets, especially on CAMS (a 7.89% improvement). However, ChatGPT$_{N\_emo}$ still underperforms ChatGPT$_{ZS}$ on most datasets, possibly because lexicon-based emotion labels are still not accurate in representing multiple emotions that co-exist in a post, especially in datasets with rich content, such as CLPsych15 and DR. Therefore, we explore the more flexible unsupervised emotion-enhanced prompts with CoT. As a result, ChatGPT$_{CoT\_emo}$ outperforms all other zero-shot methods on most datasets, which proves that emotion-enhanced CoT prompting is effective for mental health analysis. Finally, with few-shot expert-written examples, ChatGPT$_{CoT\_emo\_FS}$ significantly outperforms all zero-shot methods on all datasets, especially in complex-task datasets: t-sid 16.24% improvement, SAD 6.88% improvement, and CAMS 3.7% improvement (very approaching state-of-the-art supervised method). These encouraging results show that in-context learning is effective in calibrating LLM's decision boundaries for complex and subjective tasks in mental health analysis. We provide case studies in Appendix F.1.

| Model | Sample Num. | Avg. Token Num. | Agreement | Fluency | Reliability | Completeness | Overall |
|---|---|---|---|---|---|---|---|
| ChatGPT | 163 | 237 | 96.6% | 0.94 | 0.53 | 0.39 | 0.36 |
| ChatGPT$_{true}$ | 121 | 203 | 95.9% | 0.95 | 0.58 | 0.40 | 0.38 |
| ChatGPT$_{false}$ | 42 | 335 | 98.8% | 0.91 | 0.34 | 0.38 | 0.28 |
| InstructGPT-3 | 121 | 203 | 89.9% | 0.55 | 0.58 | 0.63 | 0.62 |

Table 2: Fleiss' Kappa and other statistics of human evaluations on ChatGPT and InstructGPT-3 results. "Sample Num." and "Avg Token Num." denote the sample numbers and average token numbers of the posts. "Agreement" denotes the percentages of results that reached a final agreement with a majority vote from the three assignments.

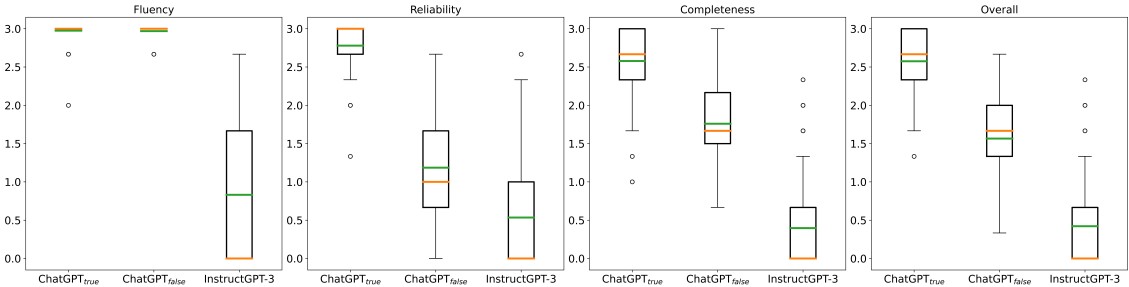

Figure 2: Box plots of the aggregated human evaluation scores for each aspect. Orange lines denote the median scores and green lines denote the average scores.

## 4.2 Evaluation Results for Explainability

**Human Evaluation**    In the above subsection, we have shown that emotion-enhanced CoT prompts can enhance ChatGPT's zero-shot performance in mental health analysis. Moreover, it can prompt LLMs to provide an explanation of their step-by-step reasoning for each response. This can significantly improve the explainability of the predictions, which is a key advantage compared with most previous black-box methods. In this subsection, we provide carefully designed human evaluations to gain a clear view of LLMs' (ChatGPT and InstructGPT-3) explainability on their detection results.

The Fleiss' Kappa results and agreement percentages are presented in Table 2. We aggregate each score by averaging assignments from three annotators, and the distributions are presented in Figure 2. Firstly, the three annotators reach high agreements on evaluation. Over 95% of ChatGPT evaluations and 89.9% of InstructGPT-3 results reach agreement. According to the widely utilized interpretation criterion [3], all Fleiss' Kappa statistics achieve at least fair agreement ($\geq$0.21) and 10 out of 16 results reach at least moderate agreement ($\geq$0.41). These outcomes prove the quality of the human annotations.

As shown in Figure 2, ChatGPT$_{true}$ almost achieves an average score of 3.0 in fluency and stably maintains outstanding performance, while

InstructGPT-3 achieves much worse performance in fluency with a 0 median score and an average score of less than 1.0. These results prove ChatGPT is a fluent explanation generator for mental health analysis. In reliability, ChatGPT$_{true}$ achieves a median score of 3 and over 2.7 in average score, showing ChatGPT as a trustworthy reasoner in supporting its classifications. Only a few of InstructGPT-3's explanations generate moderately reliable information while most of them are unreliable. For completeness, ChatGPT$_{true}$ obtains over 2.5 scores on average, indicating that ChatGPT can cover most of the relevant content in the posts to explain its classifications, while InstructGPT-3 ignores key aspects by obtaining less than 0.5 on average. Overall, ChatGPT$_{true}$ has an average score of over 2.5, proving that ChatGPT can generate human-level explanations for correct classifications regarding fluency, reliability, and completeness and significantly outperforms previous LLMs such as InstructGPT-3. More cases are in Appendix F.2.

**Automatic Evaluation**    The automatic evaluation results on the ChatGPT explanations are presented in Table 3. In ChatGPT$_{true}$, BART-Score achieves the highest correlation scores on all aspects, showing its potential in performing human-like evaluations for explainable mental health analysis. Specifically, BART-Score outperforms all BERT-Score-based methods, which shows that generative models can be more beneficial in evaluating natural

[3] https://en.wikipedia.org/wiki/Fleiss%27_kappa

| Metric | ChatGPT$_{true}$ | | | | | ChatGPT$_{false}$ | | | | |
|---|---|---|---|---|---|---|---|---|---|---|
| | Value | Fluency | Reliability | Completeness | Overall | Value | Fluency | Reliability | Completeness | Overall |
| BLEU-1 | 0.026 | 0.050 | 0.091 | 0.268 | 0.205 | 0.006 | -0.002 | 0.344 | 0.405 | 0.352 |
| ROUGE-1 | 0.226 | 0.210 | 0.258 | 0.400 | 0.351 | 0.177 | -0.010 | 0.256 | **0.539** | 0.315 |
| ROUGE-2 | 0.068 | 0.105 | 0.153 | 0.281 | 0.247 | 0.037 | 0.028 | 0.349 | 0.515 | **0.438** |
| ROUGE-L | 0.140 | 0.196 | 0.249 | 0.398 | 0.361 | 0.099 | 0.072 | 0.282 | 0.505 | 0.352 |
| GPT3-Score | -1.903 | 0.350 | 0.139 | 0.266 | 0.277 | -1.998 | -0.043 | 0.034 | 0.389 | 0.129 |
| BART-Score | -4.046 | **0.590** | **0.404** | **0.406** | **0.428** | -4.459 | -0.076 | -0.114 | 0.257 | -0.044 |
| **BERT Score-based Methods** | | | | | | | | | | |
| BERT | 0.495 | 0.080 | 0.104 | 0.224 | 0.172 | 0.470 | 0.133 | 0.192 | 0.321 | 0.180 |
| RoBERTa | 0.834 | 0.116 | 0.139 | 0.304 | 0.250 | 0.825 | 0.095 | 0.206 | 0.306 | 0.219 |
| MentalBERT | 0.552 | 0.207 | 0.214 | 0.297 | 0.263 | 0.534 | **0.148** | 0.240 | 0.255 | 0.214 |
| MentalRoBERTa | 0.763 | 0.217 | 0.177 | 0.286 | 0.260 | 0.752 | 0.129 | **0.354** | 0.427 | 0.373 |

Table 3: Pearson's correlation coefficients between human evaluation and existing automatic evaluation results on ChatGPT explanations. Best values are highlighted in bold.

language texts. Unexpectedly, BART-Score also significantly outperforms GPT3-Score, a zero-shot evaluation method based on the powerful LLM GPT-3, in all aspects. These results show that task-specific pre-training is important to trigger the language model's ability for the evaluation tasks. BART-score is also fine-tuned on text summarization and paraphrasing tasks, which are crucial to assess relevance and coherence, the two important factors for providing satisfactory evaluations. However, in ChatGPT$_{false}$, BART-Score becomes less competitive. BERT-Score achieves the best performances on fluency and reliability, and ROUGE methods outperform others on completeness and overall. A possible reason is that BERT-based methods can better distinguish false semantics in the explanations than BART. With longer posts in ChatGPT$_{false}$ (Table 2), the matching-based method ROUGE can more accurately detect the uncovered aspects in the posts, and completeness can play a more important role in determining overall performance in falsely classified samples. In BERT-Score-based methods, MentalBERT and MentalRoBERTa significantly outperform BERT and RoBERTa in most aspects, showing that pre-training on large-scale mental health texts can also benefit automatic evaluation performances of language models. More experiments on InstructGPT-3 results are presented in Appendix G.

### 4.3 Error Analysis

We further analyze some typical errors during our experiments to inspire future efforts of improving ChatGPT and emotion-enhanced prompts for mental health analysis.

**Unstable Predictions.** We notice that ChatGPT's performance on mental health analysis can vary drastically with the change of a few keywords in

| ChatGPT | DR | CLPsych15 | Dreaddit |
|---|---|---|---|
| **Zero-shot prompts** | | | |
| *any* | 82.41 | 56.31 | 53.10 |
| *some* | 74.44 | 56.59 | 50.62 |
| *very severe* | 78.65 | 47.55 | 71.79 |
| Variance | 10.6 | 17.62 | 89.29 |
| **Few-shot prompts** | | | |
| *any* | 84.74 | 57.24 | 65.62 |
| *some* | 86.9 | 61.63 | 62.0 |
| *very severe* | 84.22 | 55.19 | 75.38 |
| Variance | 1.34 | 7.21 | 31.93 |

Table 4: Change of ChatGPT's weighted-F1 performance with different adjectives with zero-shot/few-shot prompting strategies.

the prompt, especially on binary mental health condition detection. While keywords describing the tasks are easy to control, some other words such as adjectives, are hard to optimize. For example, we replace the adjective describing the mental health condition with different degrees in the zero-shot prompt for binary mental health detection:

*...Is the poster likely to suffer from [Adjective of Degree] [Condition]?...*

where the adjective (marked red) is replaced with one keyword from {*any*, *some*, *very severe*}, and the results on three binary detection datasets are shown in Table 4. As shown, ChatGPT$_{ZS}$ shows very unstable performance on all three datasets, with a high variance of 10.6 on DR, 17.62 on CLPsych15, and 89.29 on Dreaddit. There are also no global optimal adjectives as the best adjective changes with the datasets. This sensitivity makes ChatGPT's performance very unstable even with slightly different prompts. We believe this problem is due to the subjective nature of mental health conditions. The human annotations only answer

Yes/No for each post, which makes the human criteria of predictions hard to learn for ChatGPT in a zero-shot setting. To alleviate this problem, we further explore the effectiveness of few-shot prompts in these settings, where the same expert-written few-shot examples in Sec. 2.2 are included in the zero-shot prompts. As the results in Table 4 show, with few-shot prompts, ChatGPT achieves a variance of 1.34 on DR, 7.21 on CLPsych15, and 31.93 on Dreaddit, which are all significantly lower than those of zero-shot prompts. These results prove that expert-written examples can stabilize ChatGPT's predictions, because they can provide accurate references for the subjective mental health detection and cause detection tasks. The few-shot solution is also efficient as it instructs the model in an in-context learning manner, which doesn't require high-cost model fine-tuning.

**Inaccurate Reasoning.** Though ChatGPT is proven capable of generating explanations for its classifications, there are still many cases showing its inaccurate reasoning leading to incorrect results. To investigate the contributing factors behind these mistakes, we further compare the human evaluation results between the correctly and incorrectly classified results $\text{ChatGPT}_{true}$ and $\text{ChatGPT}_{false}$. The results are presented in Figure 2. As shown, $\text{ChatGPT}_{false}$ still achieves comparable fluency scores to $\text{ChatGPT}_{true}$ but performs worse on both completeness and reliability. For completeness, the average score of $\text{ChatGPT}_{false}$ drops below 2.0. We also notice that the average token number of $\text{ChatGPT}_{false}$ reaches 335 (Table 2), which exceeds $\text{ChatGPT}_{true}$ by over 130 tokens. These results indicate that ChatGPT struggles to cover all relevant aspects of long-context posts. For reliability, more than half of $\text{ChatGPT}_{false}$ results give unreliable or inconsistent explanations (below 1.0), possibly due to the lack of mental health-related knowledge. A few $\text{ChatGPT}_{false}$ samples provide mostly reliable reasoning (above 2.0) but miss key information due to the lack of completeness. Overall, the mistakes of ChatGPT are mainly caused by ignorance of relevant information in long posts and unreliable reasoning process. Therefore, future works should improve ChatGPT's long-context modeling ability and introduce more mental health-related knowledge to benefit its performance. Inaccurate reasoning also reflects a lack of alignment between LLMs and mental health analysis tasks. A possible solution is to fine-tune the LLMs with

mental health-related instruction-tuning datasets. We leave LLM-finetuning as future work. More cases are provided in Appendix F.3.

## 5 Conclusion

In this work, we comprehensively studied LLMs on zero-shot/few-shot mental health analysis and the impact of different emotion-enhanced prompts. We explored the potential of LLMs in explainable mental health analysis, by explaining their predictions via CoT prompting. We developed a reliable annotation protocol for human evaluations of LLM-generated explanations and benchmarked existing automatic evaluation metrics on the human annotations. Experiments demonstrated that mental health analysis is still challenging for LLMs, but emotional information with proper prompt engineering can better trigger their ability. Human evaluation results showed that ChatGPT can generate human-level explanations for its decisions, and current automatic evaluation metrics need further improvement to properly evaluate explainable mental health analysis. ChatGPT also bears limitations, including unstable predictions and inaccurate reasoning.

In future work, we will explore domain-specific fine-tuning for LLMs to alleviate inaccurate reasoning problems. We will also extend the interpretable settings with LLMs to other research domains.

## Acknowledgements

This work is supported by the computational shared facility at the University of Manchester and the University of Manchester President's Doctoral Scholar award. This work is supported by the project JPNP20006 from New Energy and Industrial Technology Development Organization (NEDO). Shaoxiong Ji is supported by the European Union's Horizon 2020 research and innovation program (agreement No 771113) and the EU's Horizon Europe research and innovation program under grant agreement No 101070350 and UK Research and Innovation (UKRI) under the UK government's Horizon Europe funding guarantee [grant number 10052546], also thanks the CSC - IT Center for Science, Finland for computational resources.

## Limitations

**Unexpected Responses.** Though ChatGPT makes predictions in most of its responses as

requested by the prompts, there are a few cases where it refuses to make a classification. There are two main reasons: 1) the lack of evidence from the post to make a prediction; 2) the post contains content that violates the content policy of OpenAI[4]. For example, ChatGPT can respond: "As an AI language model, I cannot accurately diagnose mental illnesses or predict what may have caused them in this post." In our experiments, we directly exclude these responses because they are very rare, but future efforts are needed to alleviate these problems.

**Limitations of Lexicons.** The motivation for using sentiment and emotion lexicons is to provide additional context with distant supervision for the prompts, which, however, have several limitations. The two lexicons, VADER (Hutto and Gilbert, 2014) and NRC EmoLex (Mohammad and Turney, 2010, 2013) we used were developed a decade ago with human annotation using social media data. It is inevitable that they suffer from annotation bias in the sentiment/emotion scores and only reflect the language used when they were developed. The Internet language evolves rapidly, and our experiments also use some recent datasets such as T-SID (Ji et al., 2022a) and CAMS (Garg et al., 2022). Besides, these lexicons have limited vocabularies. Manual rules to aggregate sentence-level sentiment and emotions could be underspecified. Prompt engineering with other advanced resources with extra emotional information can be explored in future work. We also see the limitation of the dataset. Ji (2022) showed that the sentiment distribution has no significant difference in the binary case of T-SID dataset. Although the sentiment-enhanced prompt with VADER gains slightly better performance than other prompts on T-SID dataset, we cannot clearly explain if the choice of lexicon contributes to the improvement due to the black-box nature of ChatGPT.

## Ethical Considerations

Although the datasets used are anonymously posted, our study adheres to strict privacy protocols (Benton et al., 2017; Nicholas et al., 2020) and minimizes privacy impact as much as possible, as social media datasets can reveal poster thoughts and may contain sensitive personal information. We use social posts that are manifestly

public from Reddit and Twitter. The SMS-like SAD dataset (Mauriello et al., 2021) has been released publicly on GitHub by the authors. All examples presented in our paper have been paraphrased and obfuscated using the moderate disguising scheme (Bruckman, 2002) to avoid misuse. We also do not use the user profile on social media, identify the users or interact with them. Our study aims to use social media as an early source of information to assist researchers or clinical practitioners in detecting mental health conditions for nonclinical use. The model predictions cannot replace psychiatric diagnoses. In addition, we recognize that some mental disorders are subjective (Keilp et al., 2012), and the interpretation of our analysis may differ (Puschman, 2017) because we do not understand the actual intentions of the posts.

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

# A  Related Work

## A.1  Mental Health Condition Detection

Most recently, pre-trained language models (PLMs) have been the dominant method for various NLP tasks including mental health detection. Jiang et al. (2020) proposed the attention-based user-level and post-level classification model for mental health detection with the contextual representations from BERT as input features. Zhang et al. (2022b) proposed the hierarchical attention network with BERT as the post-encoder for early depression detection. Ji et al. (2022b) released two PLMs, MentalBERT and MentalRoBERTa, for mental healthcare, which are trained with corpus from social media. Ji (2022) showcased the intention understanding capacity of PLMs via the mask prediction task and emphasized the importance of intention understanding in suicidal ideation detection. Different from the above black-box models, Han et al. (2022) proposed the explainable depression detection model hierarchical attention network (HAN), which uses the contextual embedding of BERT as input features and incorporates the metaphor concept mappings (MCMs) as the extra features to improve the interpretability of the model. Nguyen et al. (2022) proposed to improve the generalizability of BERT based depression model by grounding the detection behavior of the depression detection model to symptoms in PHQ9.

Moreover, there are also efforts incorporating multi-modal information, such as voice, video, visual, and text, to improve the performance of depression detection. Rodrigues Makiuchi et al. (2019) proposed a multi-modal method for depression detection, which incorporates speech and textual information with a gated convolutional neural network (gated CNN) and contextual feature from BERT. Lin et al. (2020) proposed the visual-textual multi-modal learning method based on CNN and BERT, for depression detection on social media. Toto et al. (2021) proposed the multi-modal method Audio-Assisted BERT (AudiBERT) for depression classification, which integrates the pre-trained audio embedding with text embedding from the bert encoder.

## A.2  Large Language Models for Mental Health Analysis

Most recently, many efforts have evaluated the performance of LLMs such as ChatGPT and GPT-4 on various NLP tasks (Bang et al., 2023; Qin

et al., 2023), such as machine translation (Jiao et al., 2023), text generation and evaluation (Benoit, 2023; Luo et al., 2023), language inference (Zhong et al., 2023). They have inspired efforts exploring the ability of LLMs for mental health analysis. Lamichhane (2023) evaluated the performance of ChatGPT on three mental health classification tasks, including stress, depression, and suicidality detection, and proved the good potential of ChatGPT for applications of mental health. Amin et al. (2023) further evaluated the capabilities of ChatGPT on big-five personality detection, sentiment analysis, and suicide detection. They show ChatGPT has better performance in sentiment analysis, comparable performance in suicide detection, and worse performance in personality detection when compared with RoBERTa-based and word embedding-based supervised methods. There are also works on analyzing the emotional reasoning ability of ChatGPT including (Qin et al., 2023; Zhong et al., 2023; Kocoń et al., 2023; Chen et al., 2023) on the sentiment classification task, where ChatGPT achieves comparable or worse performance compared with fine-tuning based methods based on PLMs. Ye et al. (2023) compared the performance of different LLMs including GPT-3 series (davinci and text-davinci-001) and GPT-3.5 series (code-davinci-002, text-davinci-002, text-davinci-003, and gpt-3.5-turbo) on the aspect-based sentiment analysis, where code-davinci-002 has the best performance in the zero-shot setting. However, most of them only cover simple binary sentiment classification tasks or a few binary mental health detection tasks, leaving a huge gap for comprehensively exploring the ability of LLMs on emotion-aware mental health analysis.

## B  ChatGPT for Emotional Reasoning

**Tasks**  We evaluate the emotional reasoning ability of ChatGPT in complex scenarios on the following two widely studied tasks: emotion recognition in conversations (ERC) and causal emotion entailment (CEE). ERC aims at recognizing the emotion of each utterance within a conversation from a fixed emotion category set, which is often modeled as a multi-class text classification task (Poria et al., 2019b). Given an utterance with a non-neutral emotion, CEE aims to identify the casual utterances for this emotion in the previous conversation history. CEE is usually modeled as a binary classification between the candidate utterance and the target utterance.

**Prompts**  We perform *direct guidance* on exploring the ability of ChatGPT in both tasks, which designs zero-shot prompts to directly ask for a classification result from the response of ChatGPT. Details about the designed prompts are presented in Appendix C.1.

**Datasets**  For ERC, we select four widely utilized benchmark datasets: IEMOCAP (Busso et al., 2008), MELD (Poria et al., 2019a), EmoryNLP (Zahiri and Choi, 2017), DailyDialog (Li et al., 2017). For CEE, we select the dataset RECCON (Poria et al., 2021). More information about these datasets is listed in Table 7 in the appendix.

**Baseline Models**  We compare the performance of ChatGPT with supervised baseline models. For ERC, we select CNN (Kim, 2014), cLSTM (Zhou et al., 2015), CNN+LSTM (Poria et al., 2017), DialogueRNN (Majumder et al., 2019), KET (Zhong et al., 2019), BERT-Base (Devlin et al., 2019), RoBERTa-Base (Liu et al., 2019), XLNet (Yang et al., 2019), DialogXL (Shen et al., 2021a), KI-Net (Xie et al., 2021), SCCL (Yang et al., 2023), and SPCL (Song et al., 2022). For CEE, we select RankCP (Wei et al., 2020), RoBERTa-Base/Large, KEC (Li et al., 2022), and KBCIN (Zhao et al., 2022). Details about these baseline models are in Appendix D.1.

**Metrics**  We use the weighted-F1 measure as the evaluation metric for IEMOCAP, MELD, and EmoryNLP datasets. Since *neutral* occupies most of DailyDialog, we use micro-F1 for this dataset, and ignore the label *neutral* when calculating the results as in the previous works (Shen et al., 2021b; Xie et al., 2021; Yang et al., 2023). For RECCON, we report the F1 scores of both negative and positive causal pairs and the macro F1 scores as a whole.

**ERC Results**  The experimental results on ERC task are presented in Table 5. We can see that ChatGPT$_{ZS}$ using the zero-shot prompting outperforms traditional supervised methods including CNN and cLSTM on IEMOCAP, MELD, and EmoryNLP datasets, showing its advantage over light-weighted supervised methods. In addition, ChatGPT$_{ZS}$ achieves comparable performance with CNN+LSTM and DialogueRNN on MELD, and EmoryNLP datasets, indicating that its

| Model | IEMOCAP | MELD | DailyDialog | EmoryNLP |
|---|---|---|---|---|
| CNN | 52.18 | 55.86 | 49.34 | 32.59 |
| cLSTM | 34.84 | 49.72 | 49.90 | 26.01 |
| CNN+LSTM | 55.87 | 56.87 | 50.24 | 32.89 |
| DialogueRNN | 61.21 | 56.27 | 50.65 | 31.70 |
| KET | 59.56 | 58.18 | 53.37 | 34.39 |
| BERT$_{Base}$ | 61.19 | 56.21 | 53.12 | 33.15 |
| RoBERTa$_{Base}$ | 55.67 | 62.75 | 55.16 | 37.0 |
| XLNet | 61.33 | 61.65 | 53.62 | 34.13 |
| DialogXL | 65.94 | 62.41 | 54.93 | 34.73 |
| KI-Net | 66.98 | 63.24 | 57.3 | – |
| SCCL | **69.88** | 65.70 | **62.51** | 38.75 |
| SPCL | 69.74 | **67.25** | – | **40.94** |
| ChatGPT$_{ZS}$ | 53.35 | 61.18 | 43.27 | 32.64 |

Table 5: Test results on ERC task. ChatGPT$_{ZS}$ denotes the method using the zero-shot prompt. The results of some baseline methods are referenced from (Zhong et al., 2019; Song et al., 2022).

generalizability can make up for the lack of task-specific model architectures to some extent. On the MELD dataset, ChatGPT$_{ZS}$ achieves 61.18% of weighted-F1 score, which outperforms some strong supervised methods including the fine-tuned BERT$_{Base}$ model (by 4.97%), and the knowledge infusion method KET (by 3.0%). However, the zero-shot performance of ChatGPT is still worse than advanced supervised methods on all datasets, and struggles to achieve dominating performance on the emotion-related tasks. This is because these tasks are very subjective even to humans, showing the promising future direction of exploring the few-shot prompting and knowledge infusion to improve the performance of ChatGPT in these subjective tasks.

| Model | Neg. F1 | Pos. F1 | Macro F1 |
|---|---|---|---|
| RankCP | **97.30** | 33.00 | 65.15 |
| RoBERTa$_{Base}$ | 88.74 | 64.28 | 76.51 |
| RoBERTa$_{Large}$ | 87.89 | 66.23 | 77.06 |
| KEC | 88.85 | 66.55 | 77.70 |
| KBCIN | 89.65 | **68.59** | **79.12** |
| ChatGPT$_{ZS}$ | 67.18 | 51.35 | 59.26 |

Table 6: Test results on the CEE task. Best values: bold. The results of baseline methods are referenced from Zhao et al. (2022).

**CEE Results** The experimental results on CEE task are presented in Table 6. We can observe that RankCP achieves the highest negative F1 score but has poor performance on the positive F1 score, which is more indicative in evaluating the emotion causal detection ability. ChatGPT$_{ZS}$ significantly outperforms RankCP on positive F1 score, showing that it possesses some level of ability to understand

the emotional causes. However, its performance is still much lower than the advanced supervised methods such as KEC and KBCIN on all metrics, which incorporate effective information such as social commonsense knowledge. Quantitatively, ChatGPT$_{ZS}$ still holds a 19.86% gap to the SOTA method KBCIN on macro F1 score.

In conclusion, the experiments on the ERC and CEE tasks show that ChatGPT holds comparable emotional reasoning ability in complex contexts with some traditional methods such as CNN and cLSTM, but still strongly underperforms competitive task-specific information infusion and fine-tuning methods. This indicates the necessity of future efforts to enhance prompting strategies and leverage external knowledge to better trigger the emotional reasoning ability of ChatGPT. These results also motivate us to design emotion-enhanced prompts to aid mental health analysis.

## C Prompt Engineering

### C.1 Emotional Reasoning

The prompt for ERC is designed as follows:

*Context: "[Previous Dialogue]". Consider this context to assign one emotion label to this utterance "[Target]". Only from this emotion list: [Emotion List]. Only return the assigned word.*

where the slots marked blue are the required inputs. *[Previous Dialogue]* denotes the previous dialogue history of the target utterance, where each utterance is pre-pended with its speaker, then concatenated in the sequence order. *[Target]* denotes the target utterance, and *[Emotion List]* denotes the predefined emotion category set of the corresponding dataset, which are listed in Table 7. Similarly, the prompt for CEE task is designed as follows:

*Context with emotion labels: "[Previous Dialogue]". Consider this context to answer the question: Did this utterance "[Query]" caused the [Target Emotion] emotion of the target utterance "[Target]"? Only return Yes or No.*

where *[Previous Dialogue]* still denotes the dialogue history with speakers, but each utterance is also post-pended with its emotion label. *[Query]* is the candidate utterance. *[Target]* and *[Target Emotion]* are the target utterance and its emotion label.

### C.2 Mental Health Analysis

**Zero-shot Prompting** We probe LLaMA, InstructionGPT-3, and ChatGPT on zero-shot

prompts. We design completion-based prompts on LLaMA, as it is not trained on instruction tuning. For example, for binary mental health condition detection, we design the following prompt:

*Post: "[Post]". The percentage that the poster is like to suffer from very severe [Condition] is*

where a predicted percentage of more than 50% is considered as positive, *[Post]* denotes the target post, *[Condition]* denotes the target mental health condition such as depression or stress, and *[List]* are the predefined labels presented in Table 8.

For InstructionGPT-3 and ChatGPT, we design instruction-based prompts since they are more natural for classification tasks. Specifically, for binary mental health condition detection, we design the following prompt:

*Post: "[Post]". Consider this post to answer the question: Is the poster likely to suffer from very severe [Condition]? Only return Yes or No.*

For multi-class mental health detection, we use the following prompt:

*Post: "[Post]". Consider this post to assign only one mental disorder label to this post from this list: [List]. Only return the assigned label.*

For cause/factor detection, the prompt is:

*Post: "[Post]". Consider this post and assign a label that causes its [Condition]. Only return answers from one of the labels: [List].*

**Emotion-enhanced CoT prompting** We design instruction-based prompts as we only test emotion-enhanced CoT prompting strategies on ChatGPT. Specifically, for the binary mental health condition detection task, we modify the zero-shot prompt as follows:

*Post: "[Post]". Consider the emotions expressed from this post to answer the question: Is the poster likely to suffer from very severe [Condition]? Only return Yes or No, then explain your reasoning step by step.*

where the green parts are the zero-shot CoT enhancements that instructs LLM to generate explanations, and the red parts are *further* added on zero-shot CoT prompts to obtain the emotion-enhanced prompts. Similar modifications are performed on the prompt of multi-class detection and cause/factor detection.

**Supervised emotion-enhanced prompting** Based on the sentiment lexicons, we assign a sentiment score to each post and convert the

score to one of the labels: *{positive, negative, neutral}*. NRC EmoLex also contains emotion annotations that were assigned from the following emotion list: *anger, anticipation, disgust, fear, joy, sadness, surprise, trust*. We regard the emotion category with the maximum emotion score as the emotion label of the input text. Specifically, VADER contains sentiment annotations with a sentiment score between -4 to 4. We utilize the NLTK package[5] to preprocess the input text (e.g., removing punctuations and singletons), apply the predefined manual rules to obtain word sentiment and aggregate the sentence-level sentiment, and return the normalized sentiment score between -1 to 1 as the overall sentiment in the post. We set the polarity threshold to 0. For NRC EmoLex, we obtain the word's stem and match the stem with the word in the lexicon when obtaining sentiment and emotion scores. Each word's emotion and sentiment scores are summed up to obtain the sentence-level sentiment and emotion. We assume there is only one sentiment or emotion in a given post and assign the post's emotion or sentiment to the categories with the maximum emotion or sentiment scores.

We design the supervised emotion-enhanced zero-shot Chain-of-Thought (CoT) prompts by adding the sentiment/emotion labels to the zero-shot prompt. For example, we modify the prompt for multi-class mental health condition detection as follows:

*Post: "[Post]". Alice thinks it is [Sentiment/Emotion]. Consider this post to assign only one mental disorder label to this post from this list: [List]. Only return the assigned label.*

where the green parts are the modifications for distantly supervised emotion infusion, and *[Sentiment/Emotion]* denotes the corresponding sentiment/emotion label. Modifications for other tasks are similar.

**Few-shot emotion-enhanced prompting** . In this section, we provide some examples of the few-shot emotion-enhanced prompts. The first example comes from DR:

*You will be presented with a post. Consider the emotions expressed in this post to answer the question: Is the poster likely to suffer from very severe depression? Only return Yes or No, then explain your reasoning step by step. Here are two examples:*

---

[5] https://www.nltk.org

*Post: Does everyone else just hurt all the time It's not like physical pain or soreness, it's just this overwhelming feeling of exhaustion and numbness...*
*Response: Yes. Reasoning: The post conveys a deep sense of emotional pain, exhaustion, and numbness. The poster describes feeling overwhelmed by even simple tasks and expresses a lack of enjoyment or fulfillment from activities that are typically pleasurable...*
*Post: Hello!: ) I'm a new user so if this post ends up in a weird place/thread, pls bear w me. Has anyone used Ancestry.com lately? They offer a two week trial...*
*Response: No. Reasoning: The post does not exhibit strong emotional indicators of very severe depression. It primarily focuses on a specific concern regarding the safety of...*

*Post: Its like that, if you want or not. ME: I have no problem, if it takes longer. But you asked my friend for help and let him wait for one hour and then you haven't prepared anything. Thats not what you asked for...*
*Response:*

Here is another example for the mental health cause detection dataset SAD:

*You will be presented post that shows the stress of the poster. Assign one label to this post only from the following stress causes list: School, Financial problem, Family issues, Social relationships, Work, Health issues, Emotional turmoil, Everyday decision making, Others. You must return the assigned labels, then explain your reasoning step by step. Here are 7 examples:*
*Post: i have been wanting to find another job for some time now*
*Response: Work. Reasoning: The post explicitly mentions that the poster has been wanting to find another job for some time now. This indicates that they are not satisfied with their current job and are experiencing stress in relation to their work situation.*
*...*
*Post: raising a teenage girl can be stressful*
*Response: Family issues. Reasoning: The post specifically mentions the task of raising a teenage girl, which falls under the category of family issues. The stress of parenting can be overwhelming, especially when dealing with a*

*sensitive age group...*

*Post: I got really scared because this happened on the way home.*
*Response:*

## D Baseline Models

### D.1 Emotion Reasoning

We select the following competitive methods for ERC:

- **CNN** (Kim, 2014) used a single-layer text-CNN to model each utterance. The classification is performed on utterance-level without contexts.

- **cLSTM** (Zhou et al., 2015) utilized a bi-directional LSTM to encode each utterance and another uni-directional LSTM to model the context.

- **CNN+LSTM** (Poria et al., 2017) used a text-CNN to model each utterance, then utilized a uni-directional LSTM to model the context based on the utterance representations.

- **DialogueRNN** (Majumder et al., 2019) used a text-CNN to extract utterance-level features, then use a separate GRU to model each participant's mental states. A global-state GRU is used to model the context.

- **KET** (Zhong et al., 2019) utilized a hierarchical Transformer to model utterances and context, and infuse word-level commonsense knowledge to enrich the semantics of the context.

- **BERT-Base** (Devlin et al., 2019) used the PLM BERT-Base to directly model the conversation. The utterance representations are used to fine-tune the weights.

- **RoBERTa-Base** (Liu et al., 2019) used a similar training paradigm to BERT-Base but with the PLM RoBERTa.

- **XLNet** (Yang et al., 2019) utilized the PLM XLNet to directly model the conversation. The segment recurrence was expected to model long contexts well.

- **DialogXL** (Shen et al., 2021a) improved the XLNet with the enhanced memory and dialog-aware self-attention mechanism to capture

| Task | Data Source | Dataset | Conv./Utter. | Emotion Category Set |
|------|-------------|---------|--------------|----------------------|
| ERC | Acted Script | IEMOCAP | 31/1,622 | *neutral, sad, anger, happy, frustrated, excited* |
| ERC | TV Show Scripts | MELD | 280/2,610 | *neutral, sad, anger, disgust, fear, happy, surprise* |
| ERC | TV Show Scripts | EmoryNLP | 85/1,328 | *neutral, sad, mad, scared, powerful, peaceful, joyful* |
| ERC | Human Written Scripts | DailyDialog | 1,000/7,740 | *neutral, happy, surprise, sad, anger, disgust, fear* |
| CEE | Human Written Scripts | RECCON | 225/2,405 | *neutral, happy, surprise, sad, anger, disgust, fear* |

Table 7: A summary of datasets for emotional reasoning in conversations. Conv. and Utter. denote conversation and utterance numbers. Data statistics are on the test set.

| Condition | Platform | Dataset | Post Num. | Labels |
|-----------|----------|---------|-----------|--------|
| Depression | Reddit | DR | 406 | *Yes, No* |
| Depression | Reddit | CLPsych15 | 300 | *Yes, No* |
| Stress | Reddit | Dreaddit | 715 | *Yes, No* |
| Suicide | Twitter | T-SID | 960 | *None, Suicide, Depression, PTSD* |
| Stress | SMS | SAD | 685 | *School, Finance, Family, Social Relation,* |
| | | | | *Work, Health, Emotion, Decision, Others* |
| Depression/Suicide | Reddit | CAMS | 626 | *None, Bias, Job, Medication, Relation, Alienation* |

Table 8: A summary of datasets for mental health tasks. Note we test the zero-shot performance on the test set.

long historical context and dependencies between multiple parties.

- **KI-Net** (Xie et al., 2021) infused both commonsense and sentiment lexicon knowledge to enhance XLNet. A self-matching module was proposed to allow interactions between utterance and knowledge representations.

- **SCCL** (Yang et al., 2023) proposed a supervised cluster-level contrastive learning (SCCL) to infuse Valance-Arousal-Dominance information. Pre-trained knowledge adapters are leveraged to incorporate linguistic and factual knowledge.

- **SPCL** (Song et al., 2022) used the PLM SimCSE (Gao et al., 2021) as the backbone model with the supervised prototypical contrastive learning (SPCL) loss.

For CEE task, we use the following baseline models:

- **RankCP** (Wei et al., 2020) ranked the clause-pair candidates in the context and utilized a neural network to perform entailment classification with the context-aware utterance representations.

- **RoBERTa-Base/Large** (Liu et al., 2019) concatenated the conversation with the emotion label of each utterance as input to the PLM RoBERTa (both RoBERTa-Base and RoBERTa-Large are used). Then CEE was modeled as a binary classification problem for each utterance pair.

- **KEC** (Li et al., 2022) utilized the directed acyclic graph networks (DAGs) incorporating social commonsense knowledge (SCK) to improve the causal reasoning ability.

- **KBCIN** (Zhao et al., 2022) proposed the knowledge-bridged causal interaction network (KBCIN) with conversational graph, emotional and actional interaction module to capture context dependencies of conversations and make emotional cause reasoning.

### D.2 Mental Health Analysis

We compare the performance of ChatGPT with that of the following baselines for mental health analysis:

- **CNN** (Kim, 2014) used three channel CNN with filters of 2,3,4 to classify the post.

- **GRU** (Cho et al., 2014) used a two-layer GRU to encode the post.

- **BiLSTM_Att** (Zhou et al., 2016) utilized a bidirectional LSTM with attention mechanism as context encoding layer to capture the contextual information of posts.

- **fastText** (Joulin et al., 2017) used an open-source and efficient text classifier based on bag of n-grams features.

- **BERT/RoBERTa** (Devlin et al., 2019; Liu et al., 2019) utilized the PLMs BERT and RoBERTa to model the post and fine-tuned them for classification.

- **MentalBERT/MentalRoBERTa** (Ji et al., 2022b) used mental healthcare-related PLMs MentalBERT and MentalRobBERTa to encode the post, which are fine-tuned for classification.

## E  Human Evaluation Criteria

Annotators will be given generated explanations from ChatGPT and InstructGPT-3, and the original post as the correct reference. Annotators will need to score and annotate the generated explanations from the following aspects:

**Fluency**  Fluency evaluates the coherence and readability of the explanation. Annotators should assess if generated explanation well-structured, easy to read, and free of grammatical or syntax errors.

- 0: Incoherent, difficult to read, and contains numerous errors

- 1: Somewhat coherent, but with poor readability and several errors

- 2: Mostly fluent, easy to read, with few minor errors

- 3: Completely fluent, coherent, and error-free

**Reliability**  Reliability measures how trustworthiness of the generated explanations to support the detection results. Annotators should assess whether the explanation is based on facts, has misinformation and wrong reasoning according to the given post. Main symptoms to check (sorted by criticality):

- Suicide ideation expressions (golden standard).

- Self-harm and self-guilt.

- Long-term low passion (e.g. loss of interest to previous hobbies).

- Loss of appetite and sleep disorders.

- Accompanied by hypersexuality or frigidity.

- Other symptoms.

The domain experts also consult other scales describing depressive symptoms, such as the Patient Health Questionnaire (PHQ-9)[6]. The annotation scheme is as follows:

- 0: Unreliable information or inconsistent information

- 1: Somewhat reliable information with some inconsistencies

- 2: Mostly reliable information with few inconsistencies

- 3: Completely reliable information

**Completeness**  Completeness measures how well the generated explanations cover all relevant aspects of the original post. Annotators should assess whether the explanation provides sufficient context and detail, without omitting important information such as emotional cues from the original post.

- 0: Omits significant information from the original post

- 1: Partially complete with some omissions

- 2: Mostly complete with minor omissions

- 3: Complete coverage of the original post

**Overall Score**  Overall performance measures the general effectiveness of the generated explanation, taking into account the combined scores for fluency, factuality, reliability, and completeness.

- 0: Poor overall performance, fluency, reliability, and completeness are all poor (=0)

- 1: Fair overall performance, fluency is good ($>= 2$), one or two aspects among reliability, and completeness is fair ($<= 1$)

- 2: Good overall performance, fluency is excellent (3), one or two aspects among reliability, and completeness is not excellent ($<= 2$)

- 3: Excellent overall performance, fluency, reliability, and completeness is all excellent (=3)

---

[6]https://www.apa.org/depression-guideline/patient-health-questionnaire.pdf

# F Case studies

## F.1 Emotion-enhanced Prompts

To provide a more intuitive view, we present an example of ChatGPT$_{CoT\_emo}$ on the mental health condition detection for a post from the Dreaddit dataset:

**Prompt**: Post: "I am scared as fuck off what the comedown is going to be like from being jacked up on adrenaline for 5 days straight. I have not felt tired whatsoever during this time..." Consider the emotion expressed from the post, then answer this question: Is the poster likely to suffer from very severe stress? You must return a Yes or No, then explain your reasoning step by step.

**Response**: Yes, the poster is likely to suffer from very severe stress. The emotional language used in the post indicates a high level of fear and anxiety, with the phrase "scared as fuck" conveying extreme distress. The fact that they have been jacked up on adrenaline for five days straight suggests that they have been in a highly stressful situation for an extended period of time... Overall, the combination of emotional and physical symptoms suggests that the poster is experiencing severe stress.

where the emotional clues and analysis are marked in red, and factual clues and analysis are marked in blue. As shown, ChatGPT with the prompt "Consider the emotion expressed from the post..." can effectively detect multiple emotions of fear, anxiety, and distress from the post and consider both clues from emotions and physical symptoms to make a final prediction. The emotion-enhanced zero-shot CoT prompt triggers ChatGPT to conduct the complex reasoning process with emotional cues, and thus improves the overall performance of mental health analysis.

## F.2 Explainability

Firstly, we provide a case for each of the binary/multi-class detection tasks. For binary detection, an example is shown in Appendix. F.1, where ChatGPT explains its reasoning process from the perspectives of emotions and physical symptoms. As shown in ChatGPT's explanation, the emotional language with the phrase "scared as fuck" and the fact "jacked up on adrenaline for 5 days straight" in the post, indicate emotional and physical symptoms of a high level of fear, anxiety, and distress.

ChatGPT thus makes the prediction of experiencing severe stress based on these symptoms.

For multi-class detection, we select a case from the T-SID dataset:

**Prompt**: Post: "Going to end my life lol see ya later haha" Consider the emotions expressed from this post, then assign only one label to this post from this mental disorders list: Depression, Suicide, PTSD, None. You must return the assigned label, then explain your reasoning step by step.

**Response**: Assigned label: Suicide. Reasoning: The post explicitly mentions the idea of ending one's life, which strongly suggests suicidal ideation. The use of humor with "lol" and "haha" may indicate a defense mechanism, such as denial or avoidance. However, given the severity of the content, it is important not to dismiss the potential for suicidal behavior. While depression and PTSD can be risk factors for suicide, they are not necessarily present in this specific post.

In this case, ChatGPT provides evidence for making a suicide ideation prediction (marked red), such as the "severity of the content" and "explicitly mentions the idea of ending one's life". It also states reasons for excluding other mental health conditions such as depression and PTSD (marked blue), such as "not necessarily present". These two-sided reasoning steps significantly improve the transparency of ChatGPT's predictions.

We provide a case from the CAMS dataset to demonstrate how ChatGPT detects the cause/factor for mental health conditions.

**Prompt**: Post: "Really have been struggling with everything lately, and I come into work today and my boss tells my I am no longer meeting expectations and fires me. I think the worst part is still being scheduled for three more days..." This post shows the depression of the poster. Consider the emotions expressed from this post and assign one label to this post only from the following depression causes list: Bias or abuse, Jobs and career, Medication, Relationship, Alienation, None. You must return the assigned label, then explain your reasoning step by step.

**Response**: Jobs and career. The post directly mentions the poster's job and their recent firing, which indicates that their job and career are a significant source of stress in their life at

*the moment. The poster also mentions struggling with everything lately, which could po-tentially be related to their job and the anxiety it has been causing them. Therefore, the label of "Jobs and career" is the most fitting for this post.*

In this case, ChatGPT extracts the direct evidence of "poster's job and their recent firing" from the post (marked blue), to support its cause detection of "Jobs and career". In addition, ChatGPT can make inferences about the poster's complaint to *potentially be related to their job* (marked green), and states their relations to the poster's anxiety emotion (marked red). These multi-aspect arguments convincingly support the final decision.

### F.3 Inaccurate Reasoning

Here is an example of the prediction from ChatGPT$_{CoT\_emo}$ for a post in the Dreaddit dataset:

*Prompt: Post: "Like she said 'shes too much attached to her boyfriend, she knows that he fucks other girls but she cant break with him cause he will block her on fb and delete her phone number forever, he is her the best friend, soulmate.' Wtf? Run away and try to fall out of love or what?" Consider the emotion expressed from the post, then answer this question: Is the poster likely to suffer from very severe stress? You must return a Yes or No, then explain your reasoning step by step.*

*Response: Yes. The fact that they are considering running away and falling out of love suggests that they may be experiencing a high level of emotional distress. The feeling of being attached to someone who is not treating them well and the fear of being blocked and deleted by that person on social media could exacerbate that distress. Therefore, it is reasonable to conclude that the poster is likely to suffer from very severe stress.*

where a poster is wrongly predicted as stressed. In this case, ChatGPT ignores the key fact that the poster was quoting the problem of another girl and felt unfair for her (marked blue). ChatGPT mistakenly analyzed the cause of stress for the mentioned girl (marked red) and put it on the poster.

## G Automatic Evaluation on InstructGPT-3

The automatic evaluation results on InstructGPT-3 explanations are presented in Table 9. The

| Metric | InstructGPT-3 | | | | |
|---|---|---|---|---|---|
| | Value | Fluency | Reliability | Completeness | Overall |
| BLEU-1 | 0.003 | 0.018 | 0.059 | 0.095 | 0.063 |
| ROUGE-1 | 0.084 | 0.401 | 0.397 | 0.490 | 0.433 |
| ROUGE-2 | 0.029 | 0.106 | 0.162 | 0.204 | 0.150 |
| ROUGE-L | 0.059 | 0.267 | 0.274 | 0.346 | 0.295 |
| GPT3-Score | -2.558 | -0.396 | -0.323 | -0.254 | -0.325 |
| BART-Score | -4.952 | 0.193 | 0.204 | 0.229 | 0.194 |
| **BERT Score-based Methods** | | | | | |
| BERT | 0.400 | **0.512** | **0.483** | **0.512** | **0.481** |
| RoBERTa | 0.814 | 0.313 | 0.288 | 0.359 | 0.315 |
| MentalBERT | 0.473 | 0.435 | 0.404 | 0.433 | 0.402 |
| MentalRoBERTa | 0.727 | 0.414 | 0.373 | 0.391 | 0.373 |

Table 9: Pearson's correlation coefficients between human evaluation and existing automatic evaluation results on InstructGPT-3 explanations. Best values are highlighted in bold.

results show that BERT-Score method based on BERT achieves the best performance in all aspects, outperforming statistics-based and generative language model-based methods. We notice that many InstructGPT-3 outputs only give diagnoses with no explanations at all, and BERT-based methods are expected to better capture the huge gap between the semantics of the original post and explanations than other methods. On the other hand, Mental-BERT and MentalRoBERTa do not outperform BERT as in ChatGPT examples, because the quality of InstructGPT-3 explanations is very low, which does not need much domain-specific knowledge to score.