# OpenReview forum: "Towards Interpretable Mental Health Analysis with Large Language Models"
_EMNLP/2023/Conference — EMNLP 2023 Main_

### Official Review · Reviewer_mk6h · 2023-08-04

**Soundness:** 4

**Excitement:**

3: Ambivalent: It has merits (e.g., it reports state-of-the-art results, the idea is nice), but there are key weaknesses (e.g., it describes incremental work), and it can significantly benefit from another round of revision. However, I won't object to accepting it if my co-reviewers champion it.

**Missing References:**

NA

**Paper Topic And Main Contributions:**

The paper studies a significant research area by examining the performance of state-of-the-art Language Model Models (LLMs), focusing on ChatGPT, in the context of mental health analysis. The authors thoroughly investigated the interpretability of these LLMs and emotional reasoning abilities across 11 datasets encompassing five distinct tasks. They studied the effectiveness of LLM's prompting strategies, explainability and explored potential limitations. To reinforce their analysis, the authors conduct extensive human and automatic evaluations, ensuring the robustness and validity of their findings. Furthermore, they contribute to the research community by releasing a new and relevant dataset in the field.



**Questions For The Authors:**

NA

**Reasons To Accept:**

1. The paper is well written.

2. The authors have addressed a crucial challenge of the effectiveness of ChatGPT and other similar LLMs in a sensitive research space like mental health.

3. The authors further exploited effective prompting methodologies such as CoT to analyze the LLM's emotional/empathetic abilities.

4. A large-scale human and automatic evaluation provide appropriate justification to assess the quality of explanations.
The proposed dataset is a significant contribution to this space.


**Reasons To Reject:**

While the paper addresses an important problem regarding the effectiveness of ChatGPT in the mental health domain, there are some concerns regarding its novelty and contribution.

1. While the paper provides an in-depth analysis of ChatGPT's performance in the mental health domain, it lacks a clear exploration of how this powerful prompt-sensitive LLM's capabilities can be effectively applied to address real-world problems or adopt stringent actions.
The paper seems to focus on understanding ChatGPT's performance rather than leveraging its potential for practical applications in mental health. Considering the remarkable abilities of such a prompt-sensitive model, it is expected that the paper should offer safe and constructive suggestions for its utilization in this domain.

2. The paper presents a comprehensive human evaluation; however, I have concerns regarding the metrics used. Fluency, Reliability, Completeness, and Overall are long-studied metrics and generally are undoubted for a) generic text generation tasks and b) models like ChatGPT. To enhance the study's rigour and provide a more in-depth understanding of the model's performance, I recommend introducing domain-centric or problem-relevant metrics that can better assess the model's suitability for mental health applications.

3. One significant concern with the paper is the lack of details about the human evaluators. It is crucial to know if these evaluators are mental health experts (Are they?) as their expertise and background can significantly impact the validity and reliability of the evaluation. Nonetheless, I strongly recommend the inclusion of an expert analysis section. This section should elaborate on the mental health experts' insights, providing substantial support and validation to the findings and may include domain-centric metrics.


**Reproducibility:**

4: Could mostly reproduce the results, but there may be some variation because of sample variance or minor variations in their interpretation of the protocol or method.

**Reviewer Confidence:**

5: Positive that my evaluation is correct. I read the paper very carefully and I am very familiar with related work.

---

> ### Author Rebuttal · Authors · 2023-08-27
>
> ## Reviewer mk6h
> We sincerely thank the reviewer for taking time to read our paper.
> We appreciate your comprehensive summarization and kind affirmation
> of this work. We
> hope the following point-to-point responses can further mitigate
> your concerns about this work. For clarity, the text in italics is
> taken verbatim from the reviewer’s comments, while the text in Roman
> font is our response.
>
> ### Reasons to reject
> *1. While the paper provides an in-depth analysis of ChatGPT's performance in the mental health domain, it lacks a clear exploration of how this powerful prompt-sensitive LLM's capabilities can be effectively applied to address real-world problems or adopt stringent actions. The paper seems to focus on understanding ChatGPT's performance rather than leveraging its potential for practical applications in mental health. Considering the remarkable abilities of such a prompt-sensitive model, it is expected that the paper should offer safe and constructive suggestions for its utilization in this domain.*
>
> Response: Thank you for pointing this out. We included less information about the potential applications of
> interpretable mental health analysis due to page limits. We present this information as follows and will include
> it in the final version of the paper:
>
> The goal of this mental health analysis model is to facilitate an understanding of a social media user's mental
> state and to promptly identify mental disorders. LLM-based models not only exhibit high levels of performance,
> but possess strong interpretability and provide insights into which features the models consider most important.
> Our model can be used for refining models and data collection strategies [1]. In addition, interpretable analysis
> can be valuable in guiding psychiatrists to better understand how certain textual features can be indicative of mental
> illness. When these professionals can understand the analysis, they are more likely to integrate these tools into their practices,
> leading to better patient-focused interventions [2].
>
> In terms of prompt-sensitivity of LLMs, we have provided a few-shot learning-based solution to effectively mitigate
> the problem.
> According to our latest experiments, few-shot in-context learning will be a promising
> solution to this problem. Expert-written examples can provide LLMs with specific criteria for subjective mental
> disorders and reduce their sensitivity to adjectives in the prompt. To further examine this hypothesis, we conduct
> experiments on the same binary mental health detection datasets in Table 4: DR, CLPsych15, and Dreaddit. Specifically, the domain experts manually write one explanation for each class in each dataset, and the explanations of each dataset are provided as context to probe ChatGPT. For example, we have the following one-shot prompt for CLPsych15:
>
> ```
> prefix: "You will be presented with a post. Consider the emotions expressed from this post, then answer the question: Does the poster show any signs of depression? You must return a Yes or No, then explain your reasoning step by step. Here are two examples:
>
> Post: RT : so gutted the arctic monkeys are headlining T when I can't go had flower of Scotland stuck in my head for three days RT : Going to the gym is always good when your in a bad mood RT : Sweet mother of Jesus RT : TWO WORLD WARS AND ONE WORLD CUP BUT YOU HAVE WON MORE WORLD CUPS &amp; YOUR ECONOMY IS BETTER THAN OURS YOU'RE PROBABLY A BETT RT : The virginity is strong with this one...
>
> Response: No. Reasoning: There are some posts that express negative emotions such as feeling gutted and missing out on events or feeling fat and unhappy. However, these types of emotions are common and do not necessarily indicate depression. Additionally, there are posts that express excitement and enjoyment for things like music concerts and dreams. Overall, the mixture of emotions expressed in the post does not strongly suggest depression.
>
> Post: : : : my life I just want Harry Styles to be my friend. I feel like he'd talk to me on my bad days. Thanks, Harry. RT : Advil PM and cough medicine that acts as a sedative...goodnight I go on tweet sprees on this account and then switch to my other one because I like it more tbh so bye My dream was in the form of a vlog so I probably need to cut down on the YouTube. My wallflower's bulb decided to...
> Response: Yes. Reasoning: He frequently mentions medication for depression, the need for therapy, feeling sad, and being in pain. Additionally, his tweeting patterns indicate that he has trouble sleeping and often feel overwhelmed. These are all common symptoms of depression, and the poster's tweets seem to reflect that they are struggling with mental health.
>
> Post:
> "
> prompt: "
> Response:"
> ```
>
> with the above prefix and prompt, the input is constructed as follows:
>
>     Input = prefix + post + prompt
>
> where ‘post’ denotes the target post. We replace the adjective “any” with “some” and “very severe” and re-prompt ChatGPT, as we did in Table 4. The new test results
> are as follows. According to the results, the variance of one-shot learning performances is much lower than those of the results in Table 4.
> This comparison shows the great potential of few-shot learning in alleviating the unstable prediction problem. We
> will include these analyses in the final version of our paper.
>
> |       ChatGPT       |  DR   | CLPsych15 | dreaddit |
> |:-------------------:|:-----:|:---------:|:--------:|
> |    *very severe*    | 84.22 |   55.19   |  75.38   |
> |       *some*        | 86.9  |   61.63   |   62.0   |
> |        *any*        | 84.74 |   57.24   |  65.62   |
> |      Variance       | 1.34  |   7.21    |  31.93   |
> | Variance in Table 4 | 10.6  |   17.62   |  89.29   |
>
>
>
> *2. The paper presents a comprehensive human evaluation; however, I have concerns regarding the metrics used. Fluency, Reliability, Completeness, and Overall are long-studied metrics and generally are undoubted for a) generic text generation tasks and b) models like ChatGPT. To enhance the study's rigour and provide a more in-depth understanding of the model's performance, I recommend introducing domain-centric or problem-relevant metrics that can better assess the model's suitability for mental health applications.*
>
> Response: As shown in Sec. 2.3 (Human evaluation), we developed our annotation scheme from two aspects.
> Since our work is the first to model mental health analysis as a generation task, we consulted human
> evaluation criteria for other text generation tasks. Secondly, we invited 2 domain experts (both are senior PhD
> students majoring in Quantitative Psychology) to further polish the scheme for mental health analysis. *Since some annotators are
> non-professional, the experts cannot develop the scheme with highly professional language.* On the other hand, all
> experts agreed that the developed scheme in appendix D is appropriate for non-professionals to evaluate the explanations. Each
> LLM-generated explanation is assigned a score by 3 annotators for each corresponding aspect. To further ensure the quality of the annotations, all annotated labels are examined by one domain expert with a domain-specific scheme, which we describe in the response below.
>
> *3. One significant concern with the paper is the lack of details about the human evaluators. It is crucial to know if these evaluators are mental health experts (Are they?) as their expertise and background can significantly impact the validity and reliability of the evaluation. Nonetheless, I strongly recommend the inclusion of an expert analysis section. This section should elaborate on the mental health experts' insights, providing substantial support and validation to the findings and may include domain-centric metrics.*
>
> Response: Following our last response, we invited two senior PhD students majoring in Quantitative Psychology as domain experts to develop the
> validation scheme and check the quality of the human annotations. The non-professional annotators mainly check the fluency and completeness of the generated explanations and give an intentional score for reliability. Then all annotated labels are examined by one domain expert.
> We will include the following expert analysis in the final version of our paper:
>
> The human annotation process follows the developed scheme in appendix D. In the annotation examination process, the domain expert mainly checks the reliability aspect. We now ask the domain experts to specify their criteria as follows:
>
> Main symptoms to check (sorted by criticality):
>
> - Suicide ideation expressions (golden standard).
> - Self-harm and self-guilt.
> - Long-term low passion (e.g. loss of interest to previous hobbies).
> - Loss of appetite and sleep disorders
> - Accompanied by hypersexuality or frigidity.
> - Other symptoms.
>
> The domain experts also consult other scales describing depressive symptoms, such as the [Patient Health Questionnaire (PHQ-9)](https://www.apa.org/depression-guideline/patient-health-questionnaire.pdf).
>
> Reliability: Domain experts check and modify the assigned reliability score with the following two steps:
>
> - Check the above symptoms in the generated explanations as support factors for classification.
> - Locate the claimed symptoms in the post to check the factuality of the explanations.
>
> ### References
> [1] Malhotra A, Jindal R. Deep learning techniques for suicide and depression detection from online social media: A scoping review[J]. Applied Soft Computing, 2022: 109713.
>
> [2] Joyce D W, Kormilitzin A, Smith K A, et al. Explainable artificial intelligence for mental health through transparency and interpretability for understandability[J]. npj Digital Medicine, 2023, 6(1): 6.

---

### Official Review · Reviewer_PbQ7 · 2023-08-04

**Soundness:** 3

**Excitement:**

3: Ambivalent: It has merits (e.g., it reports state-of-the-art results, the idea is nice), but there are key weaknesses (e.g., it describes incremental work), and it can significantly benefit from another round of revision. However, I won't object to accepting it if my co-reviewers champion it.

**Paper Topic And Main Contributions:**

This work aims to use LLMs for Mental Health Analysis. It employs LLMs to label source texts and provide explanations. It also explores different LLMs and prompting strategies. The work evaluates the LLM outputs using human and automatic methods. The human evaluation is based on a sampled dataset. The automatic evaluation is based on the source context and the LLM explanations, and it shows analysis of correlations with the human evaluation.

**Questions For The Authors:**

as for prompting:

InstructGPT3 is pretty worse in fluency and huge gaps makes one doubt if the model is called in a correct way?
Any strategies have been developed or will be developed to mitigate the above-mentioned problem in using LLMs?


as for eval: Some interesting questions that can be raised from this work are:
•  What are the factors that cause the eval inconsistency between chatGPT_true and chatGPT_false?
•  How can we use the high correlation coefficients for correct labels and low correlation coefficients for wrong labels to improve the LLM outputs?

**Reasons To Accept:**

This work demonstrates the potential of LLMs in MH analysis. It evals different LLMs and prompt strategies and human eval shows the reliability of chatGPT and the proposed emotion enhanced prompting method.

**Reasons To Reject:**

This work lacks a thorough analysis. For example, the work mentions the "unstable prediction" problem, but it does not propose/try to propose any useful solutions to mitigate it. The work attributes this problem to the "subjective nature of mental health conditions", but it does not explain in details. The work also does not address the issue of "inaccurate reasoning" either except raising the problem.

Another limitation of this work is the automatic evaluation. The work claims that the automatic metrics have a "moderate" correlation with the human evaluation, but the results in Table 3 do not support this claim. The Pearson scores are all lower than 0.5, which indicates a weak to no correlation. This suggests that the automatic metrics are not reliable or valid for evaluating the LLM outputs. However, the work does not discuss the possible reasons for this discrepancy or suggest any alternative metrics.

**Reproducibility:**

3: Could reproduce the results with some difficulty. The settings of parameters are underspecified or subjectively determined; the training/evaluation data are not widely available.

**Reviewer Confidence:**

3: Pretty sure, but there's a chance I missed something. Although I have a good feel for this area in general, I did not carefully check the paper's details, e.g., the math, experimental design, or novelty.

---

> ### Author Rebuttal · Authors · 2023-08-27
>
> ## Reviewer PbQ7
> We sincerely thank the reviewer for taking time to read our paper.
> We appreciate your comprehensive summarization and kind affirmation
> of this work. We
> hope the following point-to-point responses can further mitigate
> your concerns about this work. For clarity, the text in italics is
> taken verbatim from the reviewer’s comments while text in roman
> font is our response.
>
> ### Reasons To Reject
>
> *1. This work lacks a thorough analysis. For example, the work mentions the "unstable prediction" problem, but it does not propose/try to propose any useful solutions to mitigate it. The work attributes this problem to the "subjective nature of mental health conditions", but it does not explain in details. The work also does not address the issue of "inaccurate reasoning" either except raising the problem.*
>
> Response: For the unstable prediction problem, we have proposed to use few-shot learning as the solution, which
> mitigates the problem to a large scale.
> According to our latest experiments, few-shot in-context learning will be a promising
> solution to this problem. Expert-written examples can provide LLMs with specific criteria for subjective mental
> disorders and reduce their sensitivity to adjectives in the prompt. To further examine this hypothesis, we conduct
> experiments on the same binary mental health detection datasets in Table 4: DR, CLPsych15, and Dreaddit. Specifically, the domain experts manually write one explanation for each class in each dataset, and the explanations of each dataset are provided as context to probe ChatGPT. For example, we have the following one-shot prompt for CLPsych15:
>
> ```
> prefix: "You will be presented with a post. Consider the emotions expressed from this post, then answer the question: Does the poster show any signs of depression? You must return a Yes or No, then explain your reasoning step by step. Here are two examples:
>
> Post: RT : so gutted the arctic monkeys are headlining T when I can't go had flower of Scotland stuck in my head for three days RT : Going to the gym is always good when your in a bad mood RT : Sweet mother of Jesus RT : TWO WORLD WARS AND ONE WORLD CUP BUT YOU HAVE WON MORE WORLD CUPS &amp; YOUR ECONOMY IS BETTER THAN OURS YOU'RE PROBABLY A BETT RT : The virginity is strong with this one...
>
> Response: No. Reasoning: There are some posts that express negative emotions such as feeling gutted and missing out on events or feeling fat and unhappy. However, these types of emotions are common and do not necessarily indicate depression. Additionally, there are posts that express excitement and enjoyment for things like music concerts and dreams. Overall, the mixture of emotions expressed in the post does not strongly suggest depression.
>
> Post: : : : my life I just want Harry Styles to be my friend. I feel like he'd talk to me on my bad days. Thanks, Harry. RT : Advil PM and cough medicine that acts as a sedative...goodnight I go on tweet sprees on this account and then switch to my other one because I like it more tbh so bye My dream was in the form of a vlog so I probably need to cut down on the YouTube. My wallflower's bulb decided to make everything smell like vanilla, including my phone...
> Response: Yes. Reasoning: He frequently mentions medication for depression, the need for therapy, feeling sad, and being in pain. Additionally, his tweeting patterns indicate that he has trouble sleeping and often feel overwhelmed. These are all common symptoms of depression, and the poster's tweets seem to reflect that they are struggling with mental health.
>
> Post:
> "
> prompt: "
> Response:"
> ```
>
> with the above prefix and prompt, the input is constructed as follows:
>
>     Input = prefix + post + prompt
>
> where ‘post’ denotes the target post. We replace the adjective “any” with “some” and “very severe” and re-prompt ChatGPT, as we did in Table 4. The new test results
> are as follows. According to the results, the variance of one-shot learning performances is much lower than those of the results in Table 4.
> This comparison shows the great potential of few-shot learning in alleviating the unstable prediction problem. We
> will include these analyses in the final version of our paper.
>
> |       ChatGPT       |  DR   | CLPsych15 | dreaddit |
> |:-------------------:|:-----:|:---------:|:--------:|
> |    *very severe*    | 84.22 |   55.19   |  75.38   |
> |       *some*        | 86.9  |   61.63   |   62.0   |
> |        *any*        | 84.74 |   57.24   |  65.62   |
> |      Variance       | 1.34  |   7.21    |  31.93   |
> | Variance in Table 4 | 10.6  |   17.62   |  89.29   |
>
>
> Inaccurate reasoning reflects a lack of alignment between LLMs and mental health analysis. A possible
> solution is to fine-tune the LLMs with mental health-related instruction-tuning datasets. We leave LLM-fine-tuning
> as future work. We will include the above solutions to unstable prediction and our insights to solve inaccurate reasoning
> in the final version of our paper.
>
> *2. Another limitation of this work is the automatic evaluation. The work claims that the automatic metrics have a "moderate" correlation with the human evaluation, but the results in Table 3 do not support this claim. The Pearson scores are all lower than 0.5, which indicates a weak to no correlation. This suggests that the automatic metrics are not reliable or valid for evaluating the LLM outputs. However, the work does not discuss the possible reasons for this discrepancy or suggest any alternative metrics.*
>
> Response: Thank you for pointing out the inaccuracy. In the introduction, we mentioned “Current automatic evaluation
> metrics only moderately correlate with human evaluations”. By this, we mean the best automatic evaluation metric
> BART-score achieves moderate correlation with the human evaluation results of $ChatGPT_{true}$. We will clarify this
> sentence in the final version of our paper.
> We followed this interpretation [2] of Pearson’s Correlation Coefficient especially used for Psychology, which states coefficients between ± 0.4 and ± 0.6 as moderate correlation (Table 1 of [2]). As shown in Table 3, the BART-score achieves above 0.4 on all 4 aspects in
> results of $ChatGPT_{true}$. Therefore, we interpret BART-score as moderately correlated with the human
> evaluation results to $ChatGPT_{true}$, and can be used as a moderately reliable automatic evaluation metric for these tasks.
>
> In section 4.2 (line 503-512) we discussed
> two possible reasons for the discrepancy between BART-score and other metrics:
>
> - Generative evaluation models can be more beneficial in evaluating natural language texts [1].
> - Task-specific pre-training can be crucial for automatic evaluation models.
>
> In terms of developing task-specific automatic evaluation metrics for mental health analysis, we tried to use domain-specifically pre-trained MentalBERT and MentalRoBERTa as the base model for BERT-score, but the results were mediocre (Table 3). Therefore, we will explore better task-specific metrics to replace BART-score in future work.
>
> ### Questions For The Authors
>
> *1. InstructGPT3 is pretty worse in fluency and huge gaps makes one doubt if the model is called in a correct way? Any strategies have been developed or will be developed to mitigate the above-mentioned problem in using LLMs?*
>
> Response: We use exactly the same prompt for InstructGPT3 and ChatGPT. We are sure the model is not called in a
> wrong way. Please check gpt3_data.csv file in the supplementary materials for some examples in the binary classification
> tasks. The bad performance of InstructGPT3 in fluency is because, in our annotation process for mental health analysis,
> we specifically evaluate the fluency of the *generated explanations*. However, in many cases, InstructGPT3 cannot follow
> our CoT instructions to generate a fluent explanation for its classifications, or does not generate an explanation at all.
> In these cases, annotators also assign very low fluency scores to its response. For example, a response with no
> explanations at all: “Yes, the poster shows signs of depression.”, though fluent as a sentence, receives low fluency
> scores.
>
> The huge performance gaps between InstructGPT3 and ChatGPT mainly lie in datasets with complex tasks: t-sid, SAD,
> and CAMS. The reason is that InstructGPT3 lacks the instruction-following ability to give a meaningful response, which
> makes many of its responses unable to be evaluated.
>
> We believe the above problems mainly come from the huge gap in model sizes (InstructGPT3 has only 13B parameters)
> and lack of a high-quality instruction-tuning process. A possible strategy is to develop a high-quality instruction-tuning dataset for mental health analysis, and fine-tune InstructGPT3 on these data to align to complex mental health
> analysis tasks.
>
> *2. as for eval: Some interesting questions that can be raised from this work are: • What are the factors that cause the eval inconsistency between chatGPT_true and chatGPT_false? • How can we use the high correlation coefficients for correct labels and low correlation coefficients for wrong labels to improve the LLM outputs?*
>
> Response: For human evaluation, the eval inconsistency between $ChatGPT_{true}$ and $ChatGPT_{false}$
> mainly lie in reliability and completeness (Figure 2). For reliability, $ChatGPT_{false}$ makes incorrect
> reasoning over the post to support the false classification. For completeness, $ChatGPT_{false}$ ignored key
> facts that support the correct classification, thus making false predictions. For automatic evaluation, BART-score
> achieves moderate correlation on all aspects with human evaluation on $ChatGPT_{true}$, but performs badly on
> $ChatGPT_{false}$. The related analysis of this inconsistency is presented in Sec. 4.2 (line 513-530).
>
> The high correlation coefficients for correct labels and low correlation coefficients for wrong labels are mostly
> related to automatic evaluation methods such as the BART-score. These features are hard to be used to improve the LLMs.
> To improve the LLM outputs, we have proven few-shot learning with high-quality examples as an effective method
> (please see our responses to *Reasons to reject* 1 above
> for more details). In addition, domain-specific fine-tuning of LLMs is another promising method to improve model
> outputs, which we leave as future work.
>
> ### References
> [1] Weizhe Yuan, Graham Neubig, and Pengfei Liu. 2021. Bartscore: Evaluating generated text as text generation. Advances in Neural Information Processing Systems, 27263–27277
>
> [2] Haldun Akoglu. User's guide to correlation coefficients. In Turkish Journal of Emergency Medicine.

---

### Official Review · Reviewer_XJA8 · 2023-08-04

**Soundness:** 5

**Excitement:**

5: Transformative: This paper is likely to change its subfield or computational linguistics broadly. It should be considered for a best paper award. This paper changes the current understanding of some phenomenon, shows a widely held practice to be erroneous in someway, enables a promising direction of research for a (broad or narrow) topic, or creates an exciting new technique.

**Missing References:**

Garg, M., Shahbandegan, A., Chadha, A., & Mago, V. (2023). An Annotated Dataset for Explainable Interpersonal Risk Factors of Mental Disturbance in Social Media Posts. arXiv preprint arXiv:2305.18727.

SATHVIK, M., & Garg, M. (2023). MULTIWD: Multiple Wellness Dimensions in Social Media Posts.

**Paper Topic And Main Contributions:**

This paper addresses the application of large language models (LLMs) like ChatGPT for mental health analysis. It seeks to improve upon limitations in previous studies, such as insufficient evaluations, lack of robust prompting strategies, and lack of exploration into the explainability of LLMs.

Comprehensive Evaluation: The paper conducts a thorough assessment of mental health analysis and emotional reasoning abilities of LLMs across 11 datasets covering five tasks.

Exploring Prompting Strategies: The study investigates the effects of different prompting strategies, integrating unsupervised and distantly supervised emotional information.

Emphasizing Explainability: The authors direct LLMs to generate explanations for each of their decisions to improve the interpretability of mental health analysis.

Creating an Evaluation Dataset: A new dataset comprising 163 human-assessed explanations is created to evaluate the quality of the generated explanations. This dataset also serves as a benchmark for existing automatic evaluation metrics to guide future research.

**Questions For The Authors:**

The guidelines for human-annotated explanations in the context of mental health analysis, as described in the paper, should be carried out by domain experts due to the sensitive nature of the field. The richness of the information provided in the paper suggests that a lot of content may not have been fully explored, potentially omitting key details that would provide greater context and understanding. It appears that components from the appendix, which contain valuable information, should be integrated into the main text for a more comprehensive and coherent presentation.

**Reasons To Accept:**

One of the major strengths of this paper is its thorough evaluation of large language models (LLMs) on mental health analysis across 11 datasets and five tasks. This extensive examination offers deep insights into the capabilities of LLMs in this field, setting a solid benchmark for future work. By emphasizing the need for LLMs to generate clear explanations for their decisions, the paper tackles a critical issue in the deployment of AI systems in sensitive fields like mental health. This work could pave the way for further research on explainability in NLP models, enhancing their transparency, trustworthiness, and ethical use. The creation of a human-assessed dataset for evaluating the explanations generated by LLMs is a significant contribution. This resource can serve as a valuable tool for researchers aiming to improve and assess the explainability of their models.

**Reasons To Reject:**

While the creation of a human-assessed dataset is a significant contribution, the process of human evaluation can introduce biases and subjectivity. It's not clear from the paper how the authors mitigated these potential biases in annotation scheme or guidelines. Are the annotation guidelines expert driven or is based on some well-established documents. The study shows that ChatGPT is sensitive to minor alterations in the prompts, which might limit its utility in real-world applications. Also, there are many other datasets such as LonXplain, LOST, MultiWD, Interpersonal Risk Factors that includes identification along with the explanation/ interpretable text-spans. The author may want to consider more datasets.

**Reproducibility:**

4: Could mostly reproduce the results, but there may be some variation because of sample variance or minor variations in their interpretation of the protocol or method.

**Reviewer Confidence:**

5: Positive that my evaluation is correct. I read the paper very carefully and I am very familiar with related work.

---

> ### Author Rebuttal · Authors · 2023-08-27
>
> ## Reviewer XJA8
> We sincerely thank the reviewer for taking time to read our paper.
> We appreciate your comprehensive summarization and kind affirmation
> of this work. We
> hope the following point-to-point responses can further mitigate
> your concerns about this work. For clarity, the text in italics is
> taken verbatim from the reviewer’s comments, while the text in Roman
> font is our response.
>
> ### Reasons To Reject
>
> *1. While the creation of a human-assessed dataset is a significant contribution, the process of human evaluation can introduce biases and subjectivity. It's not clear from the paper how the authors mitigated these potential biases in annotation scheme or guidelines. Are the annotation guidelines expert driven or is based on some well-established documents.*
>
> Response: As shown in Sec. 2.3 (Human evaluation), we developed our annotation scheme from two aspects.
> Since our work is the first to model mental health analysis as a generation task, we first consulted previous human evaluation
> criteria for other text generation tasks, especially text summarization[1][2]. Secondly, we invited 2 domain experts
> (both are senior PhD students majoring in Quantitative Psychology) to further polish the scheme for mental health
> analysis. Since some
> annotators are non-professional, we asked the experts to develop the scheme in plain language, and the final scheme
> is in appendix D. Each LLM-generated explanation is assigned a score by 3 annotators for each corresponding aspect, focusing on the aspects of fluency and completeness. To further enhance the annotation quality, all annotations are examined by one domain expert. In the annotation examination process, the domain expert mainly checks the reliability aspect. We now ask the domain experts to specify their criteria as follows:
>
> Main symptoms to check (sorted by criticality):
>
> - Suicide ideation expressions (golden standard).
> - Self-harm and self-guilt.
> - Long-term low passion (e.g. loss of interest to previous hobbies).
> - Loss of appetite and sleep disorders
> - Accompanied by hypersexuality or frigidity.
> - Other symptoms.
>
> The domain experts also consult other scales describing depressive symptoms, such as the [Patient Health Questionnaire (PHQ-9)](https://www.apa.org/depression-guideline/patient-health-questionnaire.pdf).
>
> Reliability: Domain experts check and modify the assigned reliability score with the following two steps:
>
> - Check the above symptoms in the generated explanations as support factors for classification.
> - Locate the claimed symptoms in the post to check the factuality of the explanations.
>
>
> *2. The study shows that ChatGPT is sensitive to minor alterations in the prompts, which might limit its utility in real-world applications.*
>
> Response: Thank you for pointing this out. We analyzed the unstable prediction problems of LLMs in Sec. 4.3
> (unstable predictions). According to our latest experiments, few-shot in-context learning will be a promising
> solution to this problem. Expert-written examples can provide LLMs with specific criteria for subjective mental
> disorders and reduce their sensitivity to adjectives in the prompt. To further examine this hypothesis, we conduct
> experiments on the same binary mental health detection datasets in Table 4: DR, CLPsych15, and Dreaddit. Specifically, the domain experts manually write one explanation for each class in each dataset, and the explanations of each dataset are provided as context to probe ChatGPT. For example, we use the following one-shot prompt for dreaddit:
>
> ```
> prefix: "You will be presented with a post. Consider the emotions expressed from this post to answer the question: Is the poster likely to suffer from very severe stress? Only return Yes or No, then explain your reasoning step by step. Here are two examples:
>
> Post: Its like that, if you want or not. ME: I have no problem, if it takes longer. But you asked my friend for help and let him wait for one hour and then you haven’t prepared anything. Thats not what you asked for. Instead of 3 hours, he helped you for 10 hours till 5am...
> Response: No. Reasoning: The poster is expressing frustration and disappointment with someone who asked for help but did not properly prepare for it. While the situation may be stressful in the moment, there is no indication that the poster is suffering from severe or ongoing stress. The post is a specific complaint about a single incident, rather than a broader expression of emotional distress.
>
> Post: My ex used to shoot back with 'Do you want me to go with you?' all the time, no matter what it was. I almost wonder if I blocked out him asking me that about our own wedding. I unloaded a terrible memory of this on my current BF today and he took it like a champ, I just could not stop chasing myself in circles over this and knew if I heard any sort of similar phrase I would lose it. I just needed to throw this out in the open, where others who might understand could 'hear' it.
> Response: Yes. Reasoning: The poster's language reflects a high level of emotional distress and anxiety. She describes 'chasing themselves in circles', revealing that the trauma of her past experience with her ex-boyfriend continues to impact her. Additionally, the fact that she needed to share this memory with her current partner suggests that she may be struggling to cope with the aftermath of the experience. Overall, the language used and the context of the post indicates that the poster is likely suffering from severe stress.
>
> Post:
> "
>
> prompt: "
> Response:"
> ```
>
> with the above prefix and prompt, the input is constructed as follows:
>
>     Input = prefix + post + prompt
>
> where ‘post’ denotes the target post. We replace the adjective “very severe” with “some” and “any” and re-prompt ChatGPT, as we did in Table 4. The new test results
> are as follows. According to the results, the variance of one-shot learning performances is much lower than those of the results in Table 4.
> This comparison shows the great potential of few-shot learning in alleviating the unstable prediction problem. We
> will include these analyses in the final version of our paper.
>
> |       ChatGPT       |  DR   | CLPsych15 | dreaddit |
> |:-------------------:|:-----:|:---------:|:--------:|
> |    *very severe*    | 84.22 |   55.19   |  75.38   |
> |       *some*        | 86.9  |   61.63   |   62.0   |
> |        *any*        | 84.74 |   57.24   |  65.62   |
> |      Variance       | 1.34  |   7.21    |  31.93   |
> | Variance in Table 4 | 10.6  |   17.62   |  89.29   |
>
> In addition, prompt sensitivity is a common limitation of LLMs not limited to mental health detection [3][4][5]. We believe one of our main contributions is to analyze the limitations and promises (for improving explainability) of LLMs for mental health detection. It still requires much effort to make LLMs closer to realistic applications in mental health detection and also many other tasks.
>
> *3. Also, there are many other datasets such as LonXplain, LOST, MultiWD, Interpersonal Risk Factors that includes
> identification along with the explanation/ interpretable text-spans. The author may want to consider more datasets.*
>
> Response: We agree that these datasets can be very useful in examining LLMs’ mental health analysis ability. In this
> paper, we only report experiments on 11 datasets across 5 tasks due to the page limits. We are currently experimenting with these new datasets. We report zero-shot and one-shot results on MultiWD and Interpersonal Risk Factors below. Due to time limits in the rebuttal period, more results and analysis will be presented in our final version paper. Please also notice that some of the datasets are not publicly available now.
>
> |       ChatGPT              |  MultiWD   | Interpersonal Risk Factors |
> |:--------------------------:|:----------:|:--------------------------:|
> |    Zero-shot    | 62.72      |   41.33                    |
> |    One-shot    | 64.93      |   43.31                    |
>
> ### Questions For The Authors
>
> *1. The guidelines for human-annotated explanations in the context of mental health analysis, as described in the paper, should be carried out by domain experts due to the sensitive nature of the field.*
>
> Response: Our annotation scheme is developed in collaboration with two senior PhD students majoring in
> Quantitative Psychology. For more details, please refer to our response to *Reasons to reject* 1 above.
>
> *2: The richness of the information provided in the paper suggests that a lot of content may not have been fully explored, potentially omitting key details that would provide greater context and understanding. It appears that components from the appendix, which contain valuable information, should be integrated into the main text for a more comprehensive and coherent presentation.*
>
> Response: Thank you for your understanding. We moved some details to appendix because of the page limit. We will
> add more details to the methodology and experiment sections given the extra page in the final version.
>
> ### Missing References
> *Garg, M., Shahbandegan, A., Chadha, A., & Mago, V. (2023). An Annotated Dataset for Explainable Interpersonal Risk Factors of Mental Disturbance in Social Media Posts. arXiv preprint arXiv:2305.18727.*
>
> *SATHVIK, M., & Garg, M. (2023). MULTIWD: Multiple Wellness Dimensions in Social Media Posts.*
>
> Response: We will include these works in the final version of our paper.
>
> ### References
> [1] Byron C Wallace, Sayantan Saha, Frank Soboczenski, and Iain J Marshall. 2021. Generating (factual?) narrative summaries of rcts: Experiments with neural multi-document summarization. AMIA Summits on Translational Science Proceedings, 2021.
>
> [2] Jay DeYoung, Iz Beltagy, Madeleine van Zuylen, Bailey Kuehl, and Lucy Wang. 2021. Ms^2: Multi-document summarization of medical studies. In Proceedings of the 2021 Conference on Empirical Methods in Natural Language Processing, pages 7494– 752.
>
> [3] Chang et al. How to Prompt LLMs for Text-to-SQL: A Study in Zero-shot, Single-domain, and Cross-domain Settings.
>
> [4] Zheheng Luo, Qianqian Xie, and Sophia Ananiadou. 2023. Chatgpt as a factual inconsistency evaluator
> for abstractive text summarization.
>
> [5] Zhao et al. Is ChatGPT Equipped with Emotional Dialogue Capabilities?

---

### Official Review · Reviewer_NbTf · 2023-08-05

**Typos Grammar Style And Presentation Improvements:** Please review sentence in section one…
**Soundness:** 3

**Excitement:**

4: Strong: This paper deepens the understanding of some phenomenon or lowers the barriers to an existing research direction.

**Missing References:**

Zirikly, Ayah, and Mark Dredze. "Explaining models of mental health via clinically grounded auxiliary tasks. 2022
Ganesan, Adithya V., et al. "Systematic Evaluation of GPT-3 for Zero-Shot Personality Estimation.". 2023

**Paper Topic And Main Contributions:**

In this paper, the authors study the impact and use of LLM on mental health tasks (identifying mental health disorder and cause/factor detection of mental health conditions). The authors provide detailed study on how they constructed their prompts and how they enhanced the prompts to generate better explanations of why the model predicted certain condition. The output of LLM for all the tasks was automatically evaluated. Additionally, manual evaluation was conducted for the binary mental health condition classification task.
* Contributions *
- Understand the impact of prompting on mental health classification tasks
- Explore the explanations provided by LLM and conduct both human and automatic evaluation.
- Create evaluation dataset that will be publicly shared with the community.

**Questions For The Authors:**

- I really like the incorporation of the lexicons into the prompt, given the extra knowledge they provide. It is also nice to see that they improve the performance. I am a bit surprised to why ChatGPT_ZS outperforms ChatGPT_N_emo, you mention that it might be facing difficulties if there are multiple emotions in a post, however, in tweets there is usually one or two emotions that are talked about. Can you please elaborate more on this point in the paper or appendix?
- Can you compare your findings of ChatGPT + explanations to other work that investigated that outside of mental health domain? It would be interesting to know if this is specific to mental health or to ChatGPT performance in providing explanations in general.
- In Table 3, There is a big gap in correlation for reliability when comparing BART and all the other models, where reliability is really low for all the other models. Can you explain why it is low?
- You are dealing with different datasets in terms of length, content, authors (e.g. reddit post vs. tweet). Previous work such as Nguyen et al. showed that it was hard to generalize between such datasets. From your results, it seems you came across some similar findings. Do you have more insights on the difference of performance between the datasets? There are multiple factors such as imbalanced dataset, however, if you can try to balance these factors to have a more of apple-to-apple comparison, would the results across the different datasets be more close and comparable? Is this an experiment setting that you can try?
- Following the same thought, can you think of reasons where the dataset characteristics are controlling the performance of ChatGPT as opposed to the other way around?
- Can you elaborate more on the chatGPT_data.csv under human_annotation in the supplementary materials? Some of the results are yes/no and some are with explanations. I might have missed something, but wouldn't that inconsistency affects your evaluation? For instance, in chatgpt_false_data the explanations are always provided. Shouldn't it be the same with chatgpt_data.csv?

**Reasons To Accept:**

The paper is well written with a lot of details on the experiments and how the authors engineered their prompts.
The authors provide good comparison with state-of-the-art models to compare the performance against.

**Reasons To Reject:**

Thank you for your work and I enjoyed reading this paper. I found the following weaknesses that can make the paper stronger:
- The paper lacks detailed analysis about the differences between the datasets. Please check my questions below for more details.
- I think Related Work section should be part of the main paper not in appendix
- Some points that are in the main paper are vague without reading the appendix, I think this will be partially addressed when you have the extra page for the camera ready, but I think the authors should not assume that the reader needs to check the appendix to understand.
- Please check below in my questions to authors for further points that I think need to be addressed

**Reproducibility:**

3: Could reproduce the results with some difficulty. The settings of parameters are underspecified or subjectively determined; the training/evaluation data are not widely available.

**Reviewer Confidence:**

4: Quite sure. I tried to check the important points carefully. It's unlikely, though conceivable, that I missed something that should affect my ratings.

---

> ### Author Rebuttal · Authors · 2023-08-27
>
> ## Reviewer NbTf
>
> We sincerely thank the reviewer for taking time to read our paper.
> We appreciate your comprehensive summarization and kind affirmation
> of this work. We are glad that you enjoyed reading our paper, and we
> hope the following point-to-point responses can further mitigate
> your concerns about this work. For clarity, the text in italics is
> taken verbatim from the reviewer’s comments, while the text in Roman
> font is our response.
>
> ### Reasons to reject
> *1. The paper lacks detailed analysis about the differences between the datasets. Please check my questions below for more details.*
>
> Response: We analysed the influence of dataset characteristics on
> LLM performance in Sec. 4.1 (line 384-387, 403-408). We believe two characteristics are
> crucial in influencing zero-shot performances of LLMs:
>
> - the complexity of the corresponding mental health analysis task.
>
> - the gap between the dataset texts and the LLM’s training data.
>
> More details, please refer to our responses to *Questions for the authors* 4 below.
>
> *2. I think Related Work section should be part of the main paper not in appendix*
>
> Response: We moved related work to appendix due to the page limit and briefly introduced some related works in the Introduction as complementary. We will move it back to the main paper given the extra page in
> the final version. Thank you for understanding.
>
> *3. Some points that are in the main paper are vague without reading the appendix, I think this will be partially addressed when you have the extra page for the camera ready, but I think the authors should not assume that the reader needs to check the appendix to understand.*
>
> Response: Thank you again for your understanding. We moved some details of prompt designing and experimental analysis to appendix because of the page limit. With an extra page in the final version, we will enrich our methodology and experiments analysis sections to improve self-containment in the main paper.
>
> ### Questions For The Authors
>
> *1. I really like the incorporation of the lexicons into the prompt, given the extra knowledge they provide. It is also nice to see that they improve the performance. I am a bit surprised to why ChatGPT_ZS outperforms ChatGPT_N_emo, you mention that it might be facing difficulties if there are multiple emotions in a post, however, in tweets there is usually one or two emotions that are talked about. Can you please elaborate more on this point in the paper or appendix?*
>
> Response: Thank you for pointing this out. As shown in Table 1, there are four datasets (DR, CLPsych15, t-sid, and SAD) where
> the fine-grained Emolex-based method ($ChatGPT_{N\ emo}$) underperforms ChatGPT zero-shot performance
> ($ChatGPT_{ZS}$). We believe there are four main factors (we will include these analyses in the final
> version):
> - Dataset characteristics: As mentioned in Sec. 4.1 (line 437-441), lexicon-based emotion labels are
> not accurate in representing multiple emotions that co-exist in a post. The data sources (including Reddit, SMS, and Twitter) for all datasets are
> presented in Table 7, and the average token numbers of all tested datasets are provided below. As shown, DR
> and CLPsych15 from the Reddit, have long average token numbers (more than 200). Therefore, in these datasets, the posts convey
> rich content and possibly complex emotions (more than one or two emotions). In these cases, assigning a simple
> emotion label is inappropriate and can affect the model's performance.
>
> |  Dataset  | Avg. token num |Data source     |
> |:---------:|:--------------:|:--------------:|
> |    DR     |      239       |   Reddit       |
> | CLPsych15 |      1223      |   Reddit       |
> | dreaddit  |      103       |   Reddit       |
> |   t-sid   |       30       |   Twitter      |
> |    SAD    |       20       |   SMS          |
> |   CAMS    |      210       |   Reddit       |
>
>
> - Limitation of the emotion extraction methods: As shown in the paper, the lexicon emotions are extracted in a
> distantly-supervised rule-based manner, which can make mistakes in determining post emotions and handling situations
> such as sarcasm. These mistakes can be introduced as noises to the reasoning process of LLMs. On the other hand, using ChatGPT to analyse
> emotions is proven reliable in our experiments in appendix A, and other emotion-related works such as [1].
> The results in Table 1 also proved that emotion-enhanced CoT methods are more effective in leveraging emotion
> information for mental health analysis.
>
> - Relevance between emotions and mental health analysis: Though emotions have been proven to be an important factor
> in mental health analysis in most cases, it is not always effective. For example, as addressed in the Limitation
> section, [2] showed that the sentiment distribution has no significant difference in some datasets, such as T-SID
> and SAD. In these cases, only providing lexicon emotion information can have limited impact, and lead LLMs to ignore
> other important factors, such as anxiety and PTSD symptoms.
>
> - Limitations of Lexicons: This point is also addressed in the Limitation section. The two lexicons we used were
> developed a decade ago with human annotation using social media data. It is inevitable that they suffer from
> annotation bias in the sentiment/emotion scores and only reflect the language used when they were developed.
> The Internet language evolves rapidly, and the vocabularies of recently collected mental health datasets can
> bear a huge gap in the lexicons. These factors can further bring noise to the emotion extraction process.
>
> *2. Can you compare your findings of ChatGPT + explanations to other work that investigated that outside of mental health domain? It would be interesting to know if this is specific to mental health or to ChatGPT performance in providing explanations in general.*
>
> Response: ChatGPT + explanations have also been widely explored in other tasks, which also mainly leverage the chain-of-thought [4], tree-of-thought [6], or graph-of-thought [7] techniques. Successful applications include commonsense reasoning, math reasoning [4], text summarisation[5], Sudoku Puzzles[6], document Merging [7], etc. According to these results, LLMs can also provide outstanding explanations for these tasks outside of the mental health domain.
>
> Regarding our work, we did zero-shot probing on ChatGPT in the emotional reasoning domain, where the results are presented
> in appendix A. This paper focuses on exploring interpretable mental health analysis, as previous mental health analysis
> models usually modeled the tasks as black-box classification tasks, limiting their practical use in real-case
> scenarios due to their sensitive nature. Our exploration in this domain also makes this paper with rich content. We will explore LLMs+explanations in other domains in our future work.
>
> *3. In Table 3, There is a big gap in  correlation for reliability when comparing BART and all the other models, where reliability is really low for all the other models. Can you explain why it is low?*
>
> Response: In Sec. 4.2 (line 503-512), we mentioned two reasons for BART-score’s performance:
>
> - Generative evaluation models can be more beneficial in evaluating natural language texts [3].
> - Task-specific pre-training can be crucial for automatic evaluation models.
>
> These two advantages enable the BART-score to outperform other models in all aspects. In terms of reliability,
> we believe two important factors for providing satisfactory evaluation are:
>
> - Relevance: how consistent the generated hypothesis is with respect to the source text. This often requires
> evaluation models to examine if the responses correctly restate the facts in the posts.
>
> - Coherence: whether the text builds from sentence to sentence to a coherent body of information about mental
> health. This often requires evaluation models to check if the response correctly summarizes the mental states
> of the poster.
>
> Compared to GPT-score, BART-score is further fine-tuned on text summarization and paraphrasing tasks [3],
> which are highly relevant to the above two factors. Therefore, BART-score significantly outperforms other metrics
> on reliability. We will include these analyses in the final version paper.
>
> *4. You are dealing with different datasets in terms of length, content, authors (e.g. reddit post vs. tweet). Previous work such as Nguyen et al. showed that it was hard to generalize between such datasets. From your results, it seems you came across some similar findings. Do you have more insights on the difference of performance between the datasets? There are multiple factors such as imbalanced dataset, however, if you can try to balance these factors to have a more of apple-to-apple comparison, would the results across the different datasets be more close and comparable? Is this an experiment setting that you can try? Following the same thought, can you think of reasons where the dataset characteristics are controlling the performance of ChatGPT as opposed to the other way around?*
>
> Response: We agree that dataset characteristics can be a key factor influencing LLM performance. According to our experiments, two
> most important factors are (we discussed in Sec. 4.1, line 384-387, 403-408):
>
> - Complexity of the task: More complex tasks can be difficult for LLMs. For example, ChatGPT performs worse in
> multi-class mental health detection and mental health cause detection tasks than binary detection tasks. This is
> because fine-grained mental health concepts (e.g. PTSD, alienation) can be more complex and difficult for LLM to
> perceive and distinguish in a zero-shot manner. These problems can be addressed by utilizing the in-context
> learning ability of LLMs with few-shot learning.
>
> - The gap between the dataset texts and the LLM’s training data. For example, T-SID collects mostly short posts
> from Twitter with many usernames, hashtags, and slang words, which bears a huge gap with ChatGPT’s training data.
> This problem can also be alleviated by aligning LLMs with the datasets using few-shot learning or fine-tuning
> (much more costly for LLMs).
>
> However, imbalanced datasets may not be a key factor. Please note that all our experiments are done in a
> zero-shot setting, and no training processes are involved. Therefore, imbalanced data are not likely to affect the model
> performance as they normally do in supervised learning.
>
> To further evaluate the above solutions, we provide new experimental results on few-shot learning for all datasets.
> Due to resource limits, we explore one-shot learning for now. Specifically, the domain experts manually write one
> explanation for each class in each dataset, and the explanations of each dataset are provided as context to probe
> ChatGPT. For example, in DR we design the one-shot prompt as follows:
>
> ```
> prefix: "You will be presented with a post. Consider the emotions expressed from this post to answer the question: Is the poster likely to suffer from very severe depression? Only return Yes or No, then explain your reasoning step by step. Here are two examples:
>
> Post: Does everyone else just hurt all the time It’s not like physical pain or soreness, it’s just this overwhelming feeling of exhaustion and numbness. Doing anything, from applying for jobs to walking my dog, is just so overwhelmingly exhausting and it feels like running a marathon. I’ve tried everything - exercise, diet, meditation, meds...
> Response: Yes. Reasoning: The post conveys a deep sense of emotional pain, exhaustion, and numbness. The poster describes feeling overwhelmed by even simple tasks and expresses a lack of enjoyment or fulfillment from activities that are typically pleasurable. The mention of trying various methods including exercise, diet, meditation, and medication without success indicates a prolonged struggle with their emotional state. The overall tone of hopelessness, lack of positive experiences, and the feeling that every day is a struggle suggests a significant level of emotional distress that aligns with symptoms commonly associated with very severe depression.
>
> Post: Hello!: ) I'm a new user so if this post ends up in a weird place/thread, pls bear w me. Has anyone used Ancestry.com lately? They offer a two week trial but still require your credit card information. Is this suspicious? It doesn't cost anything for the first two weeks, and then you can end the trial so you don't have to pay anything even then, but I...
> Response: No. Reasoning: The post does not exhibit strong emotional indicators of very severe depression. It primarily focuses on a specific concern regarding the safety of using Ancestry.com and asking for advice and experiences from other users. While there is a slight hint of uncertainty or doubt, it does not prominently display the profound hopelessness, despair, or self-esteem issues commonly associated with very severe depression. The tone of the post seems more pragmatic and seeking information rather than expressing overwhelming emotional distress.
>
> Post: "
>
> prompt: "
> Response:"
> ```
> with the above prefix and prompt, the input is constructed as follows:
>
>     Input = prefix + post + prompt
>
> where ‘post’ denotes the target post. Following the above methods, we present the one-shot results as follows:
>
> |  Dataset  | one-shot F1 |
> |:---------:|:-----------:|
> |    DR     |    84.22    |
> | CLPsych15 |    61.63    |
> | dreaddit  |    75.38    |
> |   t-sid   |    43.95    |
> |    SAD    |    63.56    |
> |   CAMS    |    45.99    |
>
> As shown, with one-shot examples, ChatGPT significantly outperforms zero-shot results on all datasets,
> especially in complex task datasets: t-sid 16.24% improvement, SAD 6.88% improvement, and CAMS 3.7%
> improvement (very approaching SOTA performance for supervised method!). These encouraging results with simple
> one-shot learning prove that few-shot learning is an effective and efficient method to alleviate the above two problems.
> Due to resource limits, we leave few-shot learning with more examples and LLM fine-tuning as future work.
>
> *5. Can you elaborate more on the chatGPT_data.csv under human_annotation in the supplementary materials? Some of the results are yes/no and some are with explanations. I might have missed something, but wouldn't that inconsistency affects your evaluation? For instance, in chatgpt_false_data the explanations are always provided. Shouldn't it be the same with chatgpt_data.csv?*
>
> Response: It is noticed that during our human annotation, all data in chatGPT_data.csv and
> chatgpt_false_data.csv (except one or two samples) are with a Yes/No label and an explanation. They can be easily examined in our submitted datasets. We believe
> chatgpt_false_data.csv is mostly the same in format as chatgpt_data.csv. In gpt3_data.csv, most
> responses are with yes/no, but many are without explanations, because InstructGPT-3 has a huge performance gap to
> ChatGPT and cannot understand the prompt well, which leads to these inconsistencies. But these inconsistencies will
> not affect our human evaluation, because we follow the same criteria for all annotations. In cases of missing
> explanations/classifications, the response gets a very low score in fluency/reliability/completeness. This performance
> gap between InstructGPT-3 and ChatGPT is discussed in detail in Sec. 4.2 (line 474-497).
>
> ### Missing References
> *Zirikly, Ayah, and Mark Dredze. "Explaining models of mental health via clinically grounded auxiliary tasks. 2022 Ganesan, Adithya V., et al. "Systematic Evaluation of GPT-3 for Zero-Shot Personality Estimation.". 2023*
>
> Response: We will include these papers in the final version.
>
> ### References
> [1] Zhao et al. Is ChatGPT Equipped with Emotional Dialogue Capabilities?
>
> [2] Shaoxiong Ji. 2022. Towards intention understanding in suicidal risk assessment with natural language processing. In Findings of EMNLP, pages 4028–4038.
>
> [3] Weizhe Yuan, Graham Neubig, and Pengfei Liu. 2021. Bartscore: Evaluating generated text as text generation. Advances in Neural Information Processing Systems, 27263–27277
>
> [4] Jason Wei, Xuezhi Wang, Dale Schuurmans, Maarten
> Bosma, Ed Chi, Quoc Le, and Denny Zhou. 2022.
> Chain of thought prompting elicits reasoning in large
> language models.
>
> [5] Zheheng Luo, Qianqian Xie, and Sophia Ananiadou. 2023. Chatgpt as a factual inconsistency evaluator
> for abstractive text summarization.
>
> [6] Jieyi Long. Large language model guided tree-of-thought
>
> [7] Besta et al. Graph of Thoughts: Solving Elaborate Problems with Large Language Models.

---

### Meta-Review · Area_Chair_9Lh6 · 2023-09-16

**Recommendation:** 4

**Metareview:**

Authors presents a detailed analysis of performance of several state-of-the-art Large Language Models on Mental Health analysis tasks, especially mental health disorder identification and cause behind the result). Authors perform thorough evaluations under various prompting strategies, human and automated evaluations, with explanations evaluated by human experts. Most reviewers agree about the technical soundness of the paper and at least 3 found it exciting. Some reviewers found issues withe evaluations, expert choice, and lack of solutions to the cause of errors presented. Most of the comments have been address by the response.

I agree that, the "why" behind inaccurate predictions and "incorrect reasoning" (either qualitatively or quantitatively) is interesting. I will recommend authors to include this in the camera-ready.

---

### Decision · Program_Chairs · 2023-10-07

**Decision:**

Accept-Main

**Comment:**

Authors presents a detailed analysis of performance of several state-of-the-art Large Language Models on Mental Health analysis tasks, especially mental health disorder identification and cause behind the result). Authors perform thorough evaluations under various prompting strategies, human and automated evaluations, with explanations evaluated by human experts. Most reviewers agree about the technical soundness of the paper and at least 3 found it exciting. Some reviewers found issues withe evaluations, expert choice, and lack of solutions to the cause of errors presented. Most of the comments have been address by the response.

I agree that, the "why" behind inaccurate predictions and "incorrect reasoning" (either qualitatively or quantitatively) is interesting. I will recommend authors to include this in the camera-ready.